# Single-cell signaling network profiling during redox stress reveals dynamic redox regulation in immune cells

Yi-Chuan Wang [1,2], Ping-Hsun Wu[3,4,5,21], Wen-Chieh Ting[6,7,21], Yi-Fu Wang[2,21], Ming-Han Yang [2,21], Tung-Hung Su[8,21], Jia-Ying Su [9,21], Hsun-I Sun[6,7], Wei-Min Huang[7], Pei-Ling Tsai[7], Gerlinde Wernig [10], Ping-Chih Ho [11,12], Limei Wang[11,12], Chen-Tu Wu[13], Yih-Leong Chang [13], Tseng-Cheng Chen[14], Tzu-Ching Meng[15,16], Yao-Ming Chang [2], Shih-Lei Lai[2], Chia-Wei Li [2], Tai-Ming Ko [2,17], Kai-Chien Yang[2,18,19], Ya-Jen Chang [2], Yijuang Chern [2], Mei-Chuan Kuo[3,4], Yen-Tsung Huang[9], Yi-Shiuan Tzeng[2], Jih-Luh Tang[6,7,20] & Shih-Yu Chen [2] ✉

In eukaryotic cells, reactive oxygen species (ROS) serve as crucial signaling components. ROS are potentially toxic, so constant adjustments are needed to maintain cellular health. Here we describe a single-cell, mass cytometry-based method that we call signaling network under redox stress profiling (SN-ROP) to monitor dynamic changes in redox-related pathways during redox stress. SN-ROP quantifies ROS transporters, enzymes, oxidative stress products and associated signaling pathways to provide information on cellular redox regulation. Applied to diverse cell types and conditions, SN-ROP reveals unique redox patterns and dynamics including coordinated shifts in CD8[+] T cells upon antigen stimulation as well as variations in CAR-T cell persistence. Furthermore, SN-ROP analysis uncovers environmental factors such as hypoxia and T cell exhaustion for influencing redox balance, and also reveals distinct features in patients on hemodialysis. Our findings thus support the use of SN-ROP to elucidate intricate redox networks and their implications in immune cell function and disease.

Eukaryotic cells use oxygen as an energy source and must concurrently manage the reactive oxygen species (ROS) that are byproducts of the energy extraction process[1,2]. ROS are capable of reacting with various subcellular structures and can precipitate a range of cellular outcomes[3]. ROS are double-edged swords: These molecules serve as crucial mediators of signaling but are potentially toxic or stress inducing[4]. The impact of ROS on cellular destiny hinges on a delicate equilibrium between their levels and the prevailing cellular conditions. Achieving a detailed understanding of oxidative stress regulation necessitates simultaneous examination of ROS production and elimination systems as well as their downstream effectors within the spatial and temporal confines of the cell[5-7].

Cells regulate ROS levels through mechanisms that localize ROS production and by elimination through the action of antioxidant systems. ROS molecules include the superoxide anion ($O_2^{·-}$) and hydrogen peroxide ($H_2O_2$)[8]. Although $O_2^{·-}$ is generated in various cellular compartments, primarily by enzymes such as NADPH oxidases and mitochondrial electron transport chains, $O_2^{·-}$ levels are kept low due to its rapid conversion to $H_2O_2$ by superoxide dismutases (SODs), which exist in cytosolic, extracellular, and mitochondrial forms. $H_2O_2$ is generated by various oxidases in multiple subcellular locations,

including the endoplasmic reticulum and peroxisomes[9]. The decomposition of $H_2O_2$ is catalyzed by antioxidants like catalase, glutathione peroxidase, and peroxiredoxins to ensure the balance between ROS production and elimination[8]. The $H_2O_2$ balance is impacted by the transport of $H_2O_2$ across subcellular compartments through aquaporins and by activities of transcription factors, such as NRF2 and phosphorylated pNFκB (pNFκB), which induce synthesis of antioxidant defense proteins that are crucial for redox homeostasis and signaling[10].

ROS regulate vital cellular processes including growth factor signaling, the DNA damage response, stress adaptation, proliferation, and apoptosis[11]. For example, the binding of extracellular growth factors to receptor tyrosine kinases triggers a surge in ROS levels, leading to the oxidation of specific cysteine residues of protein tyrosine phosphatases, which abolishes their enzymatic activities and amplifies receptor tyrosine kinase-mediated signaling[12]. High ROS levels can harm biomolecules. Oxidative damage to DNA bases, which can lead to replication stress[11], is primarily repaired through the base excision repair pathway, initiated by a DNA glycosylase that removes damaged bases, followed by the recruitment of the apurinic/apyrimidinic endonuclease APE1[13]. Notably, APE1-mediated redox signaling facilitates binding of essential antioxidant transcription factors like pNFκB. This highlights the intricate relationships involved in redox homeostasis and emphasizes the importance of studying multiple redox pathways at the single-cell level for a detailed understanding of the redox regulation network.

Conventional proteomics techniques, such as mass spectrometry, can identify broad protein alterations resulting from ROS exposure but do not provide single-cell level resolution. Recent advancements in single-cell mass spectrometry offer a promising alternative, albeit with challenges in throughput[14]. Single-cell mass cytometry and advanced multiplexed imaging techniques have the potential to overcome these limitations. These methods can simultaneously assess intracellular conditions and surface immunophenotypes, offering a more detailed view of cellular states[15]. For instance, single-cell metabolic regulome profiling on antibody-based proteomic platforms allows quantification of the metabolic characteristics of individual cells[16,17]. Despite these advances, achieving a complete single-cell profile that encompasses the entire oxidative stress response—including the identification of ROS sources, scavengers, affected targets, and the signaling pathways involved—remains an unmet challenge.

In this study, we utilize multi-parameter, single-cell mass cytometry to map redox-associated signaling networks within individual cells. To develop the signaling network under redox stress profiling (SN-ROP) approach, we conduct a comprehensive screening of more than 100 antibodies targeting redox-related proteins to identify those suitable for single-cell profiling. The SN-ROP approach enables us to trace key redox dynamics involved in T cell activation and to identify significant alterations within the ROS network. We apply SN-ROP to analyze chimeric antigen receptor T (CAR-T) cells and other immune cells derived from patients with conditions including chronic hemodialysis and hepatocellular carcinoma, uncovering previously unrecognized signaling profiles associated with clinical outcomes or specific cellular environments. In summary, SN-ROP serves as a high-resolution platform for investigating redox-associated signaling adaptations at the single-cell level and provides new insights into immune regulation and disease pathophysiology.

## Results

### SN-ROP: A multiplexed tool for single-cell analysis of redox-associated signaling

The regulation of redox homeostasis is akin to the metabolic regulome's control over metabolic states. Both involve complex networks of regulatory elements and signaling pathways that manage production, neutralization, and cellular responses to stimuli[9]. To investigate the interconnected signaling pathways involved in redox regulation, we aimed to simultaneously quantify the abundances of ROS transporters, pivotal ROS-generating and ROS-scavenging enzymes and their regulatory modifications (e.g., phosphorylation), products of prolonged oxidative stress (e.g., sulfonic oxidation modification of proteins), and the transcription factors and signaling molecules that drive specific redox programs. Collectively, these elements form a redox-associated signaling network, which we systematically mapped using our SN-ROP platform.

To identify the most useful markers of redox-associated signaling networks, we exposed six distinct cell types, macrophage Raw264.7 cells, neuroblastoma SY5Y cells, endothelial HUVECs, cardiomyocyte HL-1 cells, Jurkat T cells, and microglial SM826 cells, to varying concentrations and durations of $H_2O_2$ treatment to simulate different ROS challenges. We evaluated 103 commercial antibodies to redox-associated factors under each of these conditions (Supplementary Table 1). By leveraging a fluorescent cell barcoding technique[18], we streamlined the analysis of 72 different experimental setups (six cell types, three $H_2O_2$ concentrations, and four time points) into a single flow cytometry assay for the equivalent of over 7,000 staining experiments (Fig. 1a and Supplementary Figs. 1a, 2a). This approach enabled the characterization of redox-regulated signaling adaptations across diverse cell types and conditions.

To select the most relevant antibodies for our SN-ROP panel, we first filtered out the antibodies that did not show any significant responses under any conditions compared to the 0-h baseline. We then grouped the remaining 72 antibodies into seven modules based on their co-regulation patterns (Supplementary Fig. 1b). Within each module, we calculated a weighted average score to assess the relative importance of each antibody and ranked them accordingly. To capture both redox and broader signaling aspects, we included eight antibodies targeting signaling pathways critical to redox balance, mTOR, HIF1α, pNFκB, phospho-S6 (pS6), c-JUN, phospho-AKT (pAKT), phospho-ERK (pERK), and phospho-p38MAPK. These antibodies were used in combination with antibodies against markers of phenotypic state to generate six panels that were used to analyze different sample types, disease models, and clinical cohorts using mass cytometry (Fig. 1b). A complete list of all signaling and phenotypic antibodies used in SN-ROP panels is provided in Supplementary Table 2. By simultaneously profiling 33 ROS-related proteins, SN-ROP captures cell-type-specific and pathway-specific redox responses (Supplementary Fig. 3 and 4), distinguishing it from traditional bulk ROS measurements.

Given the widespread use of mass spectrometry for redox status assessment, we verified our antibody-based SN-ROP mass cytometry method against this technique. We first applied SN-ROP to blood cells from ten healthy individuals and compared these findings with data from a mass spectrometry-based quantitative proteome dataset from four donors[19] (Supplementary Fig. 2b). There was a notable concordance between the SN-ROP and mass spectrometry-based datasets including a high correlation between Catalase and Ref/APE1 levels (Fig. 1c).

Next, we assessed the robustness of SN-ROP by analyzing redox network behaviors in CD8[+] T cells from OT-1 mice following antigen-specific peptide stimulation. Specifically, CytoScore, which measures the average expression of key redox markers in the cytoplasm, and MitoScore, which quantifies mitochondrial-specific redox markers, both exhibited highly correlated trends over time (Fig. 1d, Supplementary Fig. 2c). This strong correlation underscores the capability of SN-ROP to capture dynamic redox regulation at the single-cell level across different cellular compartments. Additionally, we compared the SN-ROP profiling results with previously reported RNA-seq measurements in Jurkat cells[20], which further validated the relationship between RNA and protein expression levels in response to oxidative

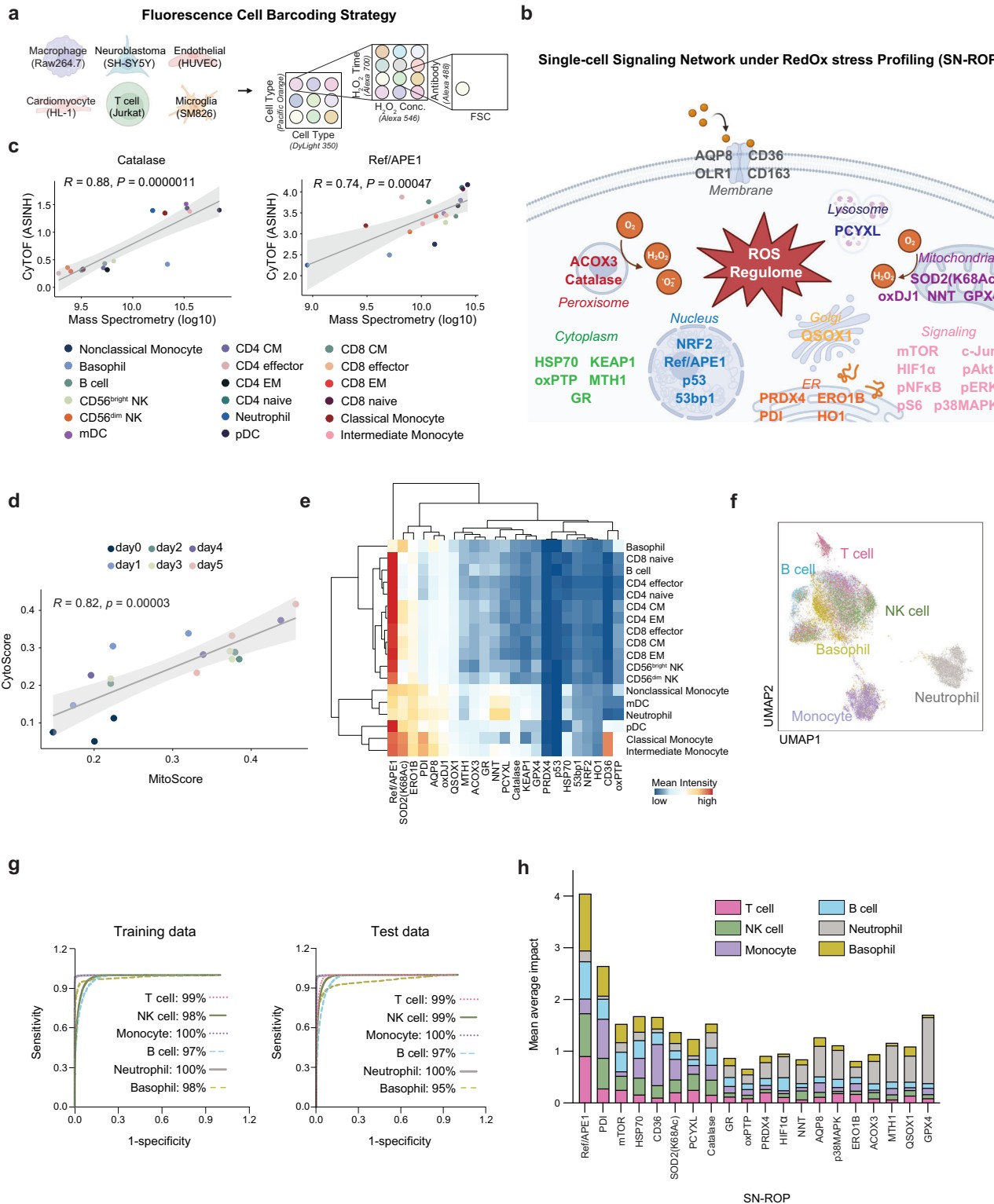

**a** Fluorescence Cell Barcoding Strategy

**b** Single-cell Signaling Network under RedOx stress Profiling (SN-ROP)

stress (Supplementary Fig. 5 and Supplementary Table 3). These observations highlight the methodological soundness of SN-ROP.

We then used SN-ROP to explore the redox profiles of various immune subpopulations under diverse conditions. First, we analyzed whole blood from healthy individuals (Supplementary Tables 4, 5). This analysis revealed that each cell type has a unique redox pattern (Fig. 1e). Specifically, markers such as Ref/APE1 are primarily associated with T and B cells, whereas NNT and PCYXL are significantly enriched in neutrophils. These findings align with our dimension reduction

analysis: The UMAP plot, based on solely redox-related features, revealed distinct segregation of the six major immune cell categories (Fig. 1f). Interestingly, we observed a group of cells composed of mixed lineages, which may be transitional cells with overlapping redox characteristics, as indicated by the differential cellular composition of effector T cells and memory T cells (Supplementary Fig. 6).

To further validate the relationship between the SN-ROP profile and immune cell phenotypes, we employed a machine learning strategy. Algorithms were trained with the SN-ROP profiles from eight

**Fig. 1 | Development and validation of the SN-ROP platform. a** Overview of the fluorescence cell barcoding strategy. Created with BioRender.com and used with permission under an Academia Sinica institutional publication license. **b** Overview of the targets of 25 ROS-related antibodies and 8 signaling-related antibodies, collectively referred to as the SN-ROP panel. Antibodies were conjugated with heavy metal isotopes for CyTOF analysis. Created with BioRender.com and used with permission under an Academia Sinica institutional publication license. **c** Pearson correlation analysis comparing marker expression in immune populations from healthy human donors determined using SN-ROP to data from a mass spectrometry-based quantitative proteome dataset[23]. Each circle represents the mean of one of the 18 immune cell subsets, distinguished by colors. CyTOF data (ASINH transformed) were collected from 10 donors, and mass spectrometry data (log10 transformed) were obtained from 4 samples. The solid line indicates the fitted linear regression; the shaded area represents the 95% confidence interval. The Pearson correlation coefficient ($R$) and exact two-sided $P$ values are indicated. **d** Pearson correlation analysis comparing CytoScore and MitoScore, which represent the overall redox states of cytoplasmic and mitochondrial compartments, respectively. Circles represent mean population values for each activated CD8+ T cells from OT-1 mice, colored by experimental day. Each data point represents triplicate measurements from CyTOF data, with ASINH-transformed expression data analyzed exclusively by mass cytometry. The solid line indicates the fitted linear regression; the shaded area represents the 95% confidence interval. The Pearson correlation coefficient ($R$) and exact two-sided $P$ values are indicated. **e** Heatmap of ASINH transformed mean expression levels of all evaluated SN-ROP markers across various immune cell lineages. **f** UMAP-based dimensionality reduction of SN-ROP data from 10 healthy donors. Colors indicate lineage identity defined by the surface markers. **g** Sensitivity versus specificity for training and test data. A subset of donors ($n = 8$) was used to train supervised machine learning algorithms to classify different immune cell types utilizing ROS markers as features. The trained models were subsequently tested on a separate set of donors ($n = 2$). **h** Mean average impacts of SN-ROP components in definition of immune cells, colored by immune cell types.

---

healthy donors, and the predictive model was tested using SN-ROP data from two additional donors. The results demonstrated prediction accuracies exceeding 95% for the six main immune subsets based on redox features only (Fig. 1g). Importantly, each marker within the SN-ROP panel played a unique role in lineage identification (Fig. 1h). For example, CD36 contributes to the definition of monocytes, and the glutathione peroxidase GPX4 is important for the definition of neutrophils. In summary, these experiments confirmed that the SN-ROP method accurately detects redox patterns associated with cell lineage.

### SN-ROP uncovers dynamic redox shifts in CD8+ T cells post-stimulation

To investigate whether T cells exhibit distinct redox regulatory networks under conditions with different functional and bioenergetic demands, we activated CD8+ T cells from OT-1 mice with antigen-specific peptides and monitored the samples over time (from day 0 to day 5) using SN-ROP. Utilizing UMAP for dimensionality reduction and focusing on redox regulators, we noted distinct separations among samples from various time points (Fig. 2a and Supplementary Table 6). As indicated by the clustering of cells from identical collection intervals, there was a shift in the redox regulatory landscape during T cell activation and dynamic changes in expression levels of all 33 SN-ROP markers (Fig. 2b).

To further elucidate the temporal dynamics of the redox changes, we employed SCORPIUS to construct a pseudotime axis that mirrors the cellular phenotypic and redox states (Fig. 2c and Supplementary Figs. 7, 8). The dynamic changes of redox-related markers were visualized as the rate of change in expression levels of each marker over pseudotime (Fig. 2d). Through this analysis, two transition points where marker modulation was coordinated were identified. The initial transition phase (pseudotime 0.3) was marked by the decrease in acetylation of SOD2 at K68 (SOD2(K68Ac)), signaling the initiation of mitochondrial dismutase activity[21]. This was accompanied by an increase in ribosomal protein pS6 and molecular chaperone HSP70, indicating that biosynthetic processes like mRNA translation and protein folding are upregulated soon after T cell stimulation. Following this phase, an increase in proteins associated with anti-oxidative redox homeostasis was observed, potentially reflecting the need for ROS scavenger activities resulting from increased ROS production that accompanies biosynthesis and bioenergetic shifts upon T cell activation[22]. Notable components of this upregulation include anti-oxidants (GPX4, Catalase), NADPH production (the nicotinamide nucleotide transhydrogenase NNT), protein folding (QSOX1), and redox signaling (p38MAPK and pNFκB). During this coordinated redox transition, expression of TCF1/7, a transcription factor that is a marker of T cell stemness[23], was elevated.

At the second transition point (pseudotime around 0.7), a reduction in anti-oxidation activities was observed, including decreases in pNFκB, the glutathione reductase GR, and GPX4 levels, suggesting a shift in the redox buffering balance. There were also upticks in the levels of oxidized protein tyrosine phosphatases (oxPTPs), which are associated with sustained oxidative stress within the cell[24]. At the second transition point, we also detected reductions in TCF1/7 and increases of EOMES and TIM3, indicative of a progression of the T cells into a terminally exhausted state. These findings highlight the dynamic interplay between redox regulation and T cell exhaustion, offering new insights into the molecular processes driving T cell dysfunction in chronic conditions.

SN-ROP analysis also revealed dynamic changes across various subcellular compartments and molecular pathways (Fig. 2e). For instance, after stimulation, there was a notable increase in the expression levels of plasma membrane receptors CD36 and OLR1, which are involved in recognizing oxidized low-density lipoproteins[25]. These levels peaked between the initial and subsequent transition phases. Similarly, Catalase and ACOX3 (two enzymes located in peroxisomes) and the nuclear DNA damage-associated molecules p53 and 53bp1 exhibited synchronized increases from the start of stimulation until the second transition point. This trend of coordinated expression was also evident in proteins engaged in the same biological functions but situated in different subcellular locales such as QSOX1, which is localized to the Golgi apparatus, and ERO1B, which is found in the endoplasmic reticulum; both these proteins are involved in disulfide bond formation[26,27]. We also observed distinct redox responses in various subcellular compartments suggestive of a sequential pattern of redox signal transduction. For example, the increase in oxidation of the mitochondrial protein DJ1 preceded the rise in oxidation of cytosolic PTP. Furthermore, there was a gradual augmentation in Ref/APE1 and its subsequent redox effector pNFκB (Fig. 2e), highlighting the complex and coordinated intracellular redox network that spans pathways and cellular compartments to maintain redox balance.

A correlation matrix across all redox regulators identified four distinct groups of features with analogous correlation patterns (Fig. 2f). Among these, a cassette enriched with protein translation and folding elements, including pS6, HSP70 and GR, the latter crucial for maintaining a reductive environment conducive to proper protein folding, was observed. This grouping also contained transcription factors TCF1/7 and Ref/APE1, suggesting a link between protein synthesis and folding with the fates of activated CD8+ T cells. Another cassette was enriched with kinase signaling, marked by upregulation of oxPTP and downstream receptor tyrosine kinases such as pERK and factors related to terminal exhaustion like EOMES and TIM3. A DNA damage and peroxidation cassette encompassing the aquaporin AQP8, Catalase, and ACOX3 as well as DNA damage marker 53bp1 and T cell activation markers CD137 and PD1 was also identified. Finally, an anti-oxidation cassette that included anti-oxidation regulators such as pNFκB, KEAP1, oxidized DJ1 (oxDJ1), and GPX4 and disulfide bond

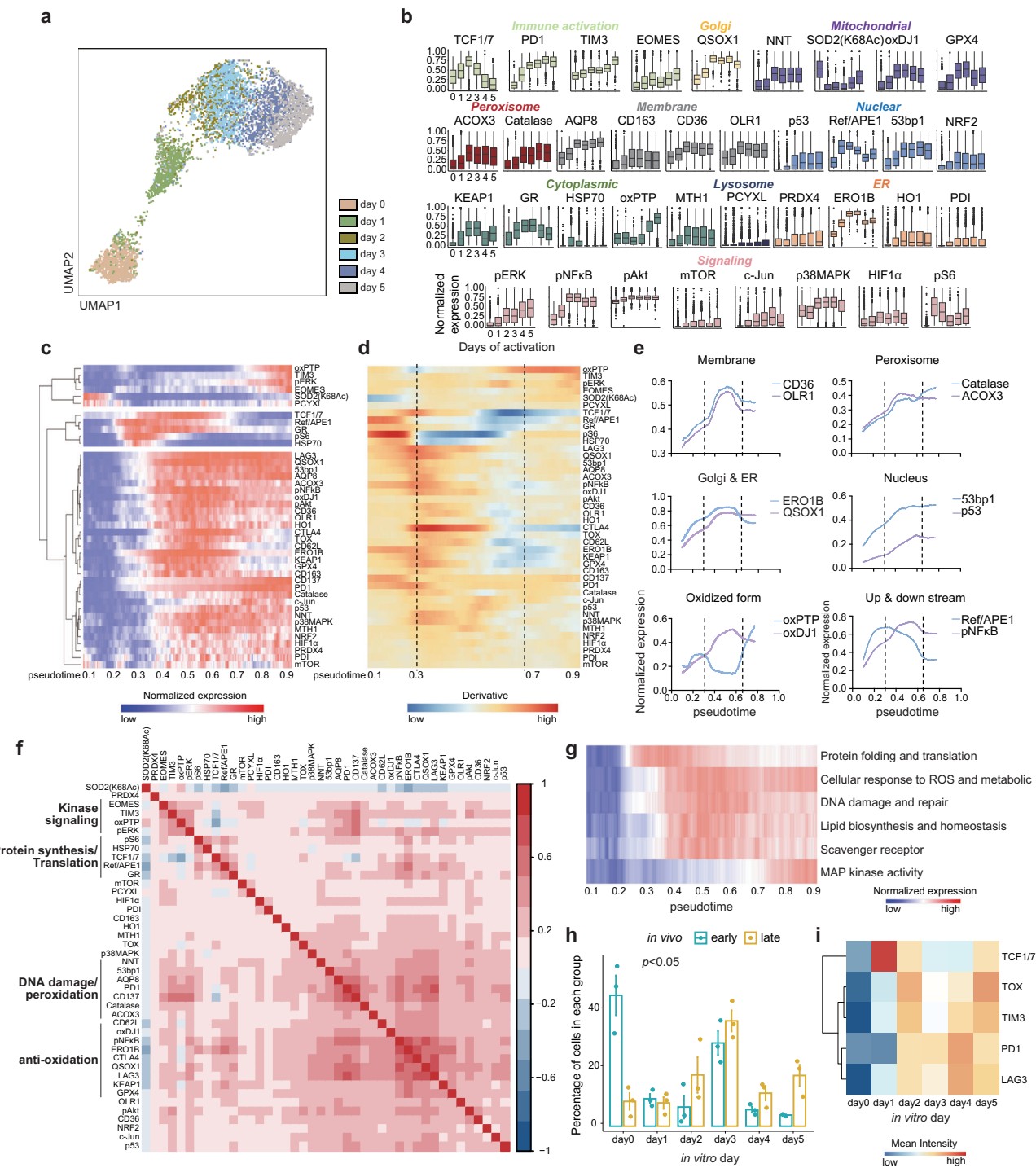

formation molecules QSOX1 and ERO1B was delineated. Intriguingly, levels of checkpoint receptors like LAG3 and CTLA4, which are activated upon persistent stimulation, were correlated with anti-oxidation modules, demonstrating the temporal integration among diverse biological processes. A plot of redox parameters along the pseudotime axis revealed the sequential activation of pathways in response to antigen stimulation (Fig. 2g).

To probe the association between redox patterns revealed by SN-ROP analysis of CD8⁺ T cells activated in vitro with in vivo data, we adopted a machine learning framework to devise a model capable of inferring in vivo activation extent from in vitro redox signatures. As an in vivo system, we analyzed CD8⁺ T cells from the MC38 colorectal cancer mouse model using SN-ROP

(Supplementary Fig. 2d and Supplementary Table 7). CD8⁺ T cells isolated 7 days after tumor introduction closely matched the early activation phases predicted by the model, whereas cells collected at the 14-day mark were aligned with more advanced stages of activation (Fig. 2h). We also observed significant increases in the expression of late-stage T cell activation markers such as TOX, TIM3, PD1, and LAG3 and a decrease in TCF1/7 expression in cells categorized into later activation stages (Fig. 2i). These findings suggest that the dynamic shifts in the redox landscape observed in vitro faithfully replicate those observed in CD8⁺ T cells in vivo. These results highlight the effectiveness of the SN-ROP technique in revealing dynamic and coordinated redox molecular details at the single-cell level upon T cell activation.

**Fig. 2 | SN-ROP reveals remodeling of ROS signaling networks in CD8⁺ T cells.**
**a** UMAP projection of SN-ROP from pooled CD8⁺ T cells from OT-I mice at days 0-5 of OVA stimulation colored to show the distribution of cells across different time points. **b** Box plots displaying 99th percentile normalized SN-ROP expression profiles for each CD8⁺ T cell activation time point ($n = 3$ independent samples). Each box represents the distribution of single-cell expression values. Box plots show the median (center line), interquartile range (IQR; box limits), and whiskers extending to 1.5×IQR; outliers beyond this range are shown as individual points. The plots are colored based on protein function or subcellular localization. No statistical comparisons were performed. **c** Pseudotime values calculated using the SCORPIUS package plotted in a heatmap along with the 99th percentile normalized data, which was smoothed using a window size of 100 ($n = 3$ independent samples, data from one representative sample shown). **d** Slope (first derivative) heatmap of protein expression across pseudotime. The vertical dashed lines indicate significant inflection points ($n = 3$ independent samples, data from one representative sample shown). **e** Examples of expression of SN-ROP markers in organelles as a function of pseudotime. Data shown are the 99th percentile normalized values for a representative sample ($n = 3$) smoothed using a window size of 1000. The vertical dashed

lines indicate inflection times. **f** ASINH transformed data of CD8⁺ T cells from OT-I mice annotated with GO biological processes. Red represents a positive correlation, and blue represents a negative correlation ($n = 3$ independent samples, data from one representative sample shown). The features are grouped into four categories based on their functional roles: kinase signaling (EOMES, TIM3, oxPTP, pERK); protein synthesis/translation (HSP70, TCF1/7, REF/APE1, and GR); DNA damage/peroxidation (NNT, 53bp1, AQP8, PD1, CD137, Catalase, and ACOX3); and antioxidation (CD62L, oxDJI, pNFκB, ERO1B, CTLA4, QSOX1, LAG3, KEAP1, and GPX4). **g** Pseudotime heat map with biological processes of the ROS functions divided into six pathways based on membership in one or more functional GO modules or pathways. **h** The percentage of CD8⁺ T cells from MC38 tumors at in vivo day 7 (early, blue) and day 14 (late, gold) distributed across in vitro time points ($n = 3$ biologically independent MC38 tumor-bearing mice per group). Bars represent the mean, and error bars indicate ± standard error of the mean (SEM). Statistical significance was assessed using a two-sided permutation t-test. **i** Mean expression levels of five late-stage activation markers in CD8⁺ T cells from MC38 tumors projected onto the in vitro timeline ($n = 3$).

## Redox shifts revealed by SN-ROP are correlated with CAR-T cell persistence

To verify that the dynamic and coordinated redox responses in T cells upon activation in mice revealed by SN-ROP analysis are relevant to humans, we conducted SN-ROP analyses of samples taken from seven patients with CD19⁺ lymphoid leukemia who were undergoing CAR-T therapy (Supplementary Tables 8, 9). Peripheral blood samples collected at multiple time points (0, 7, 14, 21, 28, and 90 days after treatment) were analyzed (Fig. 3a and Supplementary Fig. 2e). Our analysis focused on CAR-positive T cells. These cells were categorized into 15 clusters based on their redox profiles using FlowSOM (Supplementary Fig. 9). Two distinct types of redox responses were observed: an active type characterized by high redox activity as indicated by markers such as oxPTP, PDI, and Ref/APE1 (clusters 1, 2, 3, 4, 5, 6, 8, 11, and 13) and a basal type with low redox activity (clusters 7, 9, 10, 12, 14, and 15) (Fig. 3b, c and Supplementary Figs. 9, 10).

Further investigation into the temporal dynamics between these clusters and the proportion of CAR-positive T cells revealed two highly correlated patterns, distinguished by their dominance in either the active or basal subsets as a function of time after treatment initiation (Fig. 3d). Intriguingly, the percentage of CAR-positive T cells at day 90 was predominantly associated with the basal subset module. An analysis of active-type CAR-positive T cells over time for each patient showed that a significant portion of CAR-T cells in patients were initially the active type but that this proportion decreased over time in those with higher frequencies of CAR-positive T cells at day 90 (Fig. 4a). Two clusters classified as basal type showed strong correlations with CAR-positive T cells at day 90 (clusters 9 and 12: $R^2 = 0.9171$ and 0.8713, respectively); in contrast, active-type clusters exhibited negative correlations (clusters 3 and 5: $R^2 = 0.692$ and 0.8214, respectively) (Fig. 4b and Supplementary Fig. 11). PDI and ERO1B were expressed at low levels in T cells at day 28 post-infusion in patients with high CAR-T persistence (e.g., patient 1908) and at high levels in patients with low CAR-T persistence (e.g., patient 1903) (Fig. 4c). These findings demonstrate that there is dynamic and coordinated regulation of redox responses post antigen-specific activation in T cells from both humans and mice. More importantly, our method revealed a strong correlation between temporal fluctuations in redox activity and in vivo CAR-T persistence, suggesting the potential of redox activity as a biomarker.

## Environmental perturbations influence redox patterns and T cell exhaustion

In tumor environments, oxygen availability crucially influences redox states, which in turn affect the functionality of immune cells[28]. To

explore how oxygen levels impact cellular redox dynamics, we activated CD8⁺ T cells from OT-1 mice under 20% oxygen (normoxia) and 1.5% oxygen (hypoxia). We collected cells after short (day 2) and long (day 4) periods of hypoxia and analyzed samples with SN-ROP and with pimonidazole staining as a proxy for cellular hypoxia levels (Supplementary Figs. 2c, 12). The consistency of our SN-ROP analysis was underscored by the reproducible patterns observed across multiple biological replicates (Supplementary Fig. 13). A comparative redox profile analysis between hypoxically and normoxically cultured CD8⁺ T cells highlighted the pronounced impact of reduced oxygen tension on redox balance, especially after long hypoxic exposure (Fig. 5a, b and Supplementary Figs. 14, 15 and Supplementary Table 10). Critical antioxidants like peroxiredoxin PRDX4, glutathione peroxidase GPX4, and key transcription factors NRF2 and pNFκB were increased in response to hypoxic stress (Fig. 5a). Despite upregulation of these factors, the rise in oxidative stress indicators like oxPTP and the diminished functional activity of SOD2(K68Ac) indicated that redox buffering capacity was compromised under hypoxic stress (Fig. 5a–c). Furthermore, these redox adjustments were correlated with typical signs of T cell exhaustion, including elevated PD1 and TIM3 and reduced TCF1/7 expression (Fig. 5c).

Pathway analysis revealed a gradual intensification in the cellular response to ROS as hypoxia was prolonged, whereas the protein folding and translation pathways were increased at day 2 of hypoxia but were similar at day 4 in normoxic and hypoxic conditions (Fig. 5c, d). These results imply that there is a predisposition in the redox response of CD8⁺ T cells toward antioxidation that likely serves as a compensatory mechanism against persistent hypoxic stress and the advancement of T cell exhaustion. We applied conditional Density-Rescaled Visualization (DREVI) to map the interaction between hypoxia levels and redox or T cell differentiation signals (Supplementary Fig. 16). The conditional density plot identified a distinct hypoxic threshold beyond which there was a coordinated increase in antioxidant and T cell exhaustion markers (Fig. 5e). In contrast, markers associated with protein folding, translation (HSP70 and pS6), and progenitor cell status (TCF1/7) were suppressed upon hypoxic stress (Fig. 5e).

To determine if these redox pathways are similarly modulated by hypoxia in human CD8⁺ T cells in vivo, we examined dissociated CD8⁺ T cells from the border, core, and unaffected areas of tumors in two patients with hepatocellular carcinoma (Supplementary Fig. 2f and Supplementary Tables 11, 12). Hypoxic stress was significant in the cancer core region as indicated by elevated HIF1α expression compared to that in the junction and normal regions (Fig. 5f). In addition, T cells from the tumor core mirrored the redox response of hypoxia-exposed mouse CD8⁺ T cells, but cells from the tumor border and

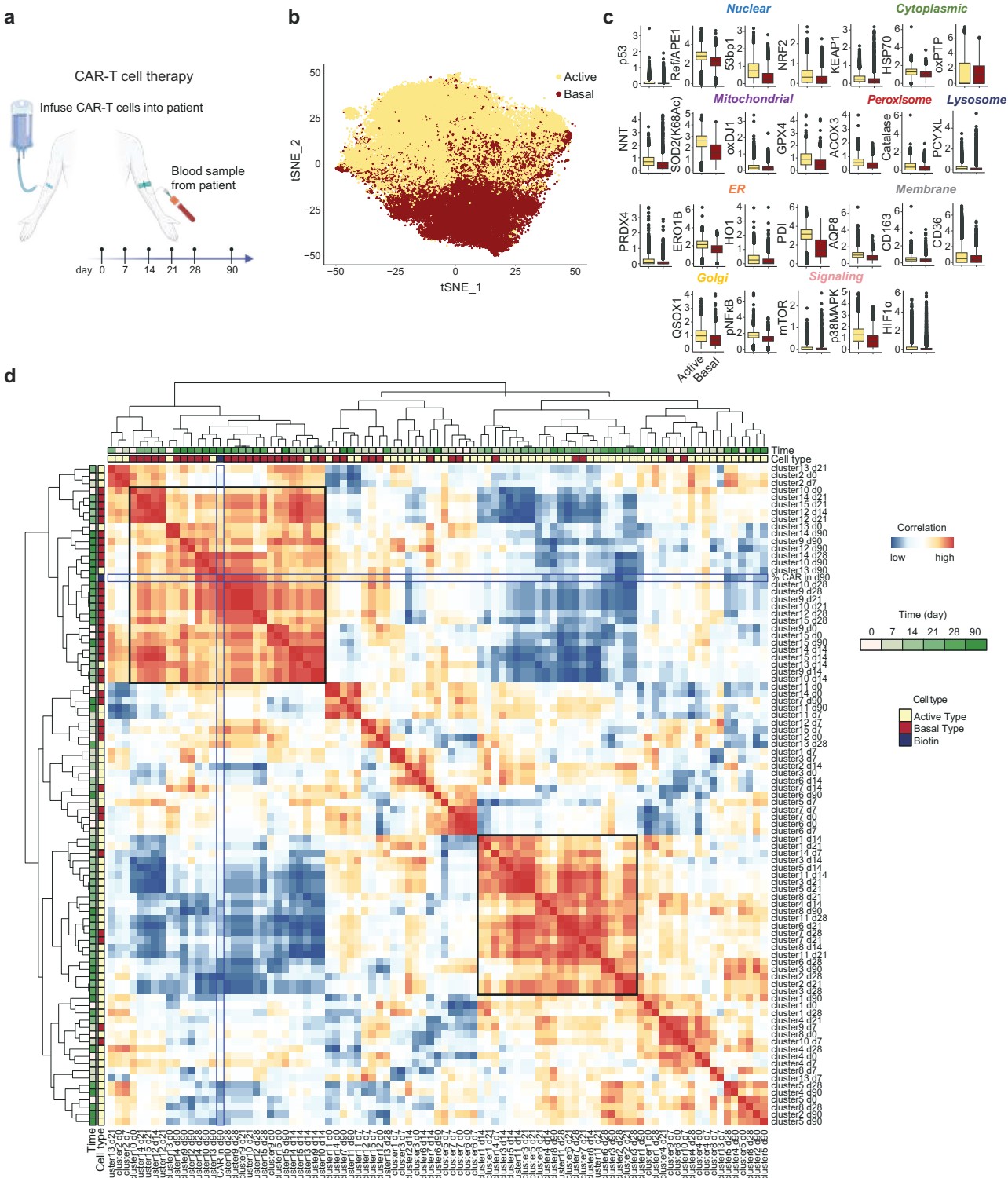

**Fig. 3 | Early SN-ROP-defined redox patterns distinguish CAR-T cell states across time. a** Diagram of the experimental protocol. Blood was sampled from leukemia patients (*n* = 7) undergoing CAR-T therapy on days 0, 7, 14, 21, 28, and 90 after CAR-T infusion. CD8⁺ T cells were subsequently isolated and analyzed using SN-ROP. Created with BioRender.com and used with permission under an Academia Sinica institutional publication license. **b** tSNE plot of CAR-positive T cells, clustered based on their redox profiles using FlowSOM. **c** ASINH-transformed expression levels of SN-ROP markers for active (yellow) and basal (red) clusters. Each box represents the distribution of single-cell expression values derived from 7 biologically independent CAR-T cell patient samples. Box plots display the median (center line), interquartile range (IQR; box limits), and whiskers extending to 1.5×IQR. Minima and maxima beyond the whiskers are shown as individual points (outliers). No statistical comparisons were performed. **d** Correlation heatmap of all clusters across all sample collection time points. Upper square marks the correlated basal type and the lower right square denotes the active type detected at the 90-day time point. Blue squares highlight biotin⁺ (i.e., CAR-T-positive) cells at 90 days post-infusion.

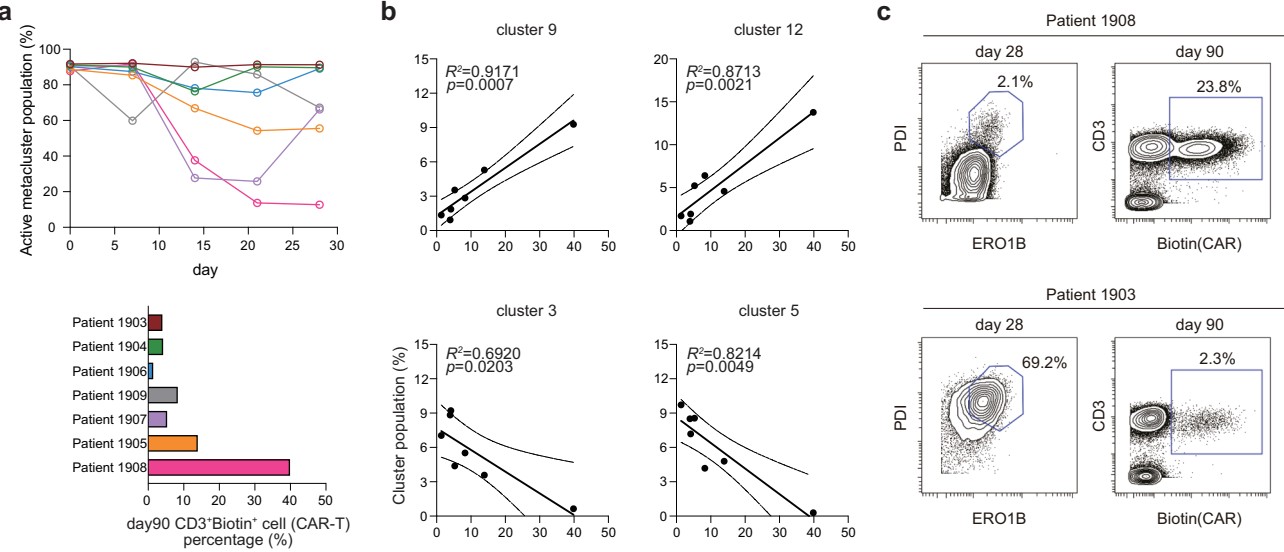

**Fig. 4 | Persistent SN-ROP activation patterns correlate with long-term CAR-T cell persistence. a** Upper: Percentages of active-type T cell clusters as a function of time in individual patients. Lower: Percentages of biotin⁺ cells at 90 days post-infusion in each patient. **b** Correlations of the percentages of indicated cell clusters with CAR-T cell persistence at day 90 post-infusion. Each dot represents an individual CAR-T patient. The solid line represents the fitted linear regression. Pearson correlation coefficients ($R^2$) and exact two-sided $P$ values are shown in the plots. Results for additional clusters are provided in Supplementary Fig. 11. **c** Contour plots of PDI versus ERO1B expression at day 28 post-infusion (left) and CAR-T cells versus all T cells (right) in samples from patients 1908 and 1903 at day 90 post-infusion.

normal regions did not (Fig. 5f). This pattern included the increased expression of antioxidant molecules such as Catalase and NRF2 as well as signs of inadequate redox buffering including SOD2 acetylation and increased oxPTP. In sum, our investigations of both murine models and human hepatocellular carcinoma samples revealed that distinct redox shifts in CD8⁺ T cells are triggered by hypoxic environmental conditions.

Recent studies have revealed that modulating intracellular ROS levels in T cells can significantly impact the progression of T cell exhaustion[29,30]. To explore the cellular mechanisms that influence this effect during T cell activation, we activated T cells from OT-1 mice in vitro with or without N-acetylcysteine (N-AC), an antioxidant known to decrease intracellular ROS levels[31], and conducted SN-ROP. Under hypoxia, redox alterations were most apparent in late in the time course, but N-AC treatment caused significant changes immediately (Fig. 6a, b and Supplementary Fig. 2c, 17). On day 1, we observed elevated expression of molecules related to protein translation and folding (pS6, ERO1B, and QSOX1) and of components of antioxidant system such as GPX4, KEAP1, and pNFκB (Fig. 6b), suggesting that without the antioxidant reinforcement from N-AC, these redox-sensitive pathways are intensified to counteract the ROS generated upon T cell activation. The immediate impact of N-AC treatment was further corroborated by pathway analyses, which revealed upregulation in nearly all redox-related pathways by day 1, particularly those related to anti-oxidation and protein translation and folding (Fig. 6c). Interestingly, despite the reduced redox differences between control and N-AC-treated groups later in the time course, the continuous suppression of exhaustion markers and oxidation byproducts (oxPTP) along with a noticeable increase in the stemness markers (TCF1/7) in the N-AC-treated group persisted throughout the time course (Fig. 6b, d).

To validate these findings in a physiological context, we utilized the well-established in vivo T cell exhaustion model of chronic lymphocytic choriomeningitis virus (LCMV) infection. In the acute LCMV model, mice are infected with the LCMV Armstrong strain, leading to a robust and transient immune response[32]. The virus is cleared rapidly, within 7-10 days, and T cell activation levels return to baseline. The chronic LCMV model utilizes the LCMV clone 13 strain, which induces a prolonged and persistent infection that results in sustained T cell activation and eventual exhaustion[32]. SN-ROP analyses of samples from both models indicated that chronic infection was characterized by elevated oxidative byproducts (oxPTP), diminished functional activity of SOD2(K68Ac), increased exhaustion markers (PD1 and TIM3), and reduced stemness markers (TCF1/7) over time (Fig. 6d, Supplementary Figs. 2d, 18, and Supplementary Table 13). These results are similar to those observed in vitro.

To further elucidate the causal link between the early redox equilibrium and T cell effector functions, we analyzed the effector responses of in vitro antigen stimulated T cells from OT-1 mice in the presence of APX2009, an inhibitor targeting the Ref/APE1 pathway, which acts upstream of pNFκB, a key regulator of antioxidant mechanisms[33](Supplementary Fig. 2g). We observed a significant decrease in the proportion of TNFα- and IFNγ-producing CD8⁺ T cells in the APX2009 treatment group compared to the controls (Fig. 5e). Importantly, the addition of exogenous N-AC counteracted this reduction (Fig. 5e), demonstrating that the effects of APX2009 on T cell functionality are redox dependent.

To gain deeper mechanistic insights into how Ref/APE1-mediated early redox equilibrium impacts T cell exhaustion, we applied the SN-ROP platform to compare APX2009-treated cells, N-AC-treated cells, and cells treated with the combination. APX2009 treatment disrupted key redox correlation networks as we observed the loss of connections between Ref/APE1-ERO1B and GPX4-ERO1B-pNFκB (Supplementary Fig. 19). Interestingly, these disrupted links were restored by the addition of N-AC, suggesting that APX2009 interferes with the coordination of redox signaling, particularly interactions between the endoplasmic reticulum and mitochondrial components such as GPX4. Consistent with this, mitochondrial fitness, measured using Mito-Tracker Deep Red, was reduced following APX2009 treatment (Fig. 5f), indicating that disrupting Ref/APE1-mediated redox equilibrium impairs mitochondrial function and drives T cell exhaustion. These findings underscore the indispensable role of a coordinated redox response in T cell function and affirm the value of investigating the redox response through advanced single-cell analysis techniques.

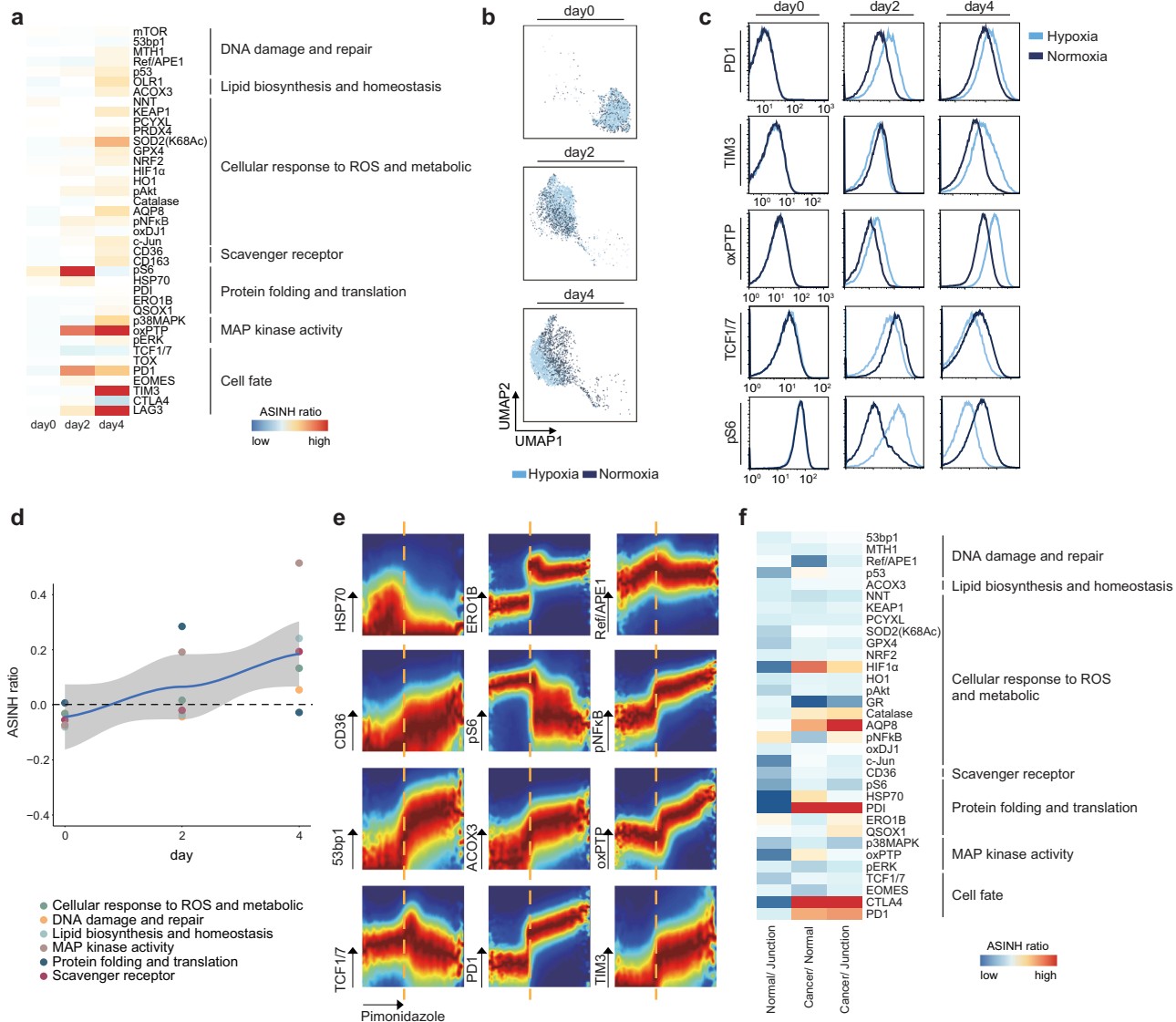

**Fig. 5 | SN-ROP analysis of CD8⁺ T cells under normal and hypoxic conditions reveals correlations between redox patterns and T cell exhaustion. a** ASINH ratio heatmap of SN-ROP marker expression in CD8⁺ T cells from OT-1 mice cultured in hypoxic versus normoxic conditions. The markers are grouped by six GO pathway terms. Results show the average effect sizes across replicates ($n = 3$ per condition and time point). **b** UMAP of individual time points for CD8⁺ T cells under normoxic and hypoxic conditions. The input features used are from the SN-ROP markers only, and the plots are colored to distinguish different conditions. Plotted is a representative result of 2000 cells from one of the triplicate experiments. **c** Histograms of exhaustion and SN-ROP markers at days 0, 2, and 4. The histograms are colored to indicate normoxia or hypoxia and are from one of the triplicate experiments. **d** Loess scatter plot of ASINH ratios in hypoxia over normoxia

conditions in a representative sample. Each point represents a different GO term. The black dashed line represents ASINH ratio of 0. The solid line represents a locally weighted regression (LOESS) fit; the shaded area indicates the 95% confidence interval around the fitted curve. **e** DREVI plot with the density estimate renormalized to visualize the abundance of the SN-ROP markers under hypoxia (as observed by pimonidazole). Each plot depicts the distribution of densities, with dark red indicating higher density within that specific slice. The orange dashed line indicates the coordinated transition hypoxic time point. **f** Heatmap of ratios of expression patterns of the SN-ROP markers in CD8⁺ T cells in normal versus junction regions, cancerous regions versus normal regions, and cancerous regions versus junction regions from two hepatocellular carcinoma patients. ASINH ratios are grouped by GO term.

## SN-ROP reveals distinct redox features in hemodialysis patients

To further demonstrate the applicability of the SN-ROP approach under disease conditions, we obtained peripheral blood samples from hemodialysis patients ($n = 33$) and age-matched healthy controls ($n = 6$) (Supplementary Tables 5, 14) and employed the SN-ROP for simultaneous redox state evaluation and immunophenotyping. Utilizing phenotypic marker-based manual gating, we delineated 18 unique immune cell populations (Supplementary Fig. 2b). Examination of the distribution of these immune subsets between healthy subjects and those undergoing hemodialysis revealed that only B cells differed significantly (Fig. 7a and Supplementary Fig. 20). However, a detailed analysis of

redox markers for each immune subset, totaling 453 cell-type-specific redox attributes, identified 36 redox features that significantly separated healthy participants from hemodialysis patients following adjustments for multiple comparisons (Fig. 7b). We used these significant redox features to re-cluster the individuals and compared this to the immunophenotyping only. No distinction was observed through immunophenotyping alone, whereas SN-ROP correctly clustered healthy and diseased subjects (Fig. 7c, d). This analysis revealed that despite the similarity in peripheral blood cell populations between healthy individuals and hemodialysis patients, the underlying cellular redox pattern is significantly altered in patients undergoing hemodialysis.

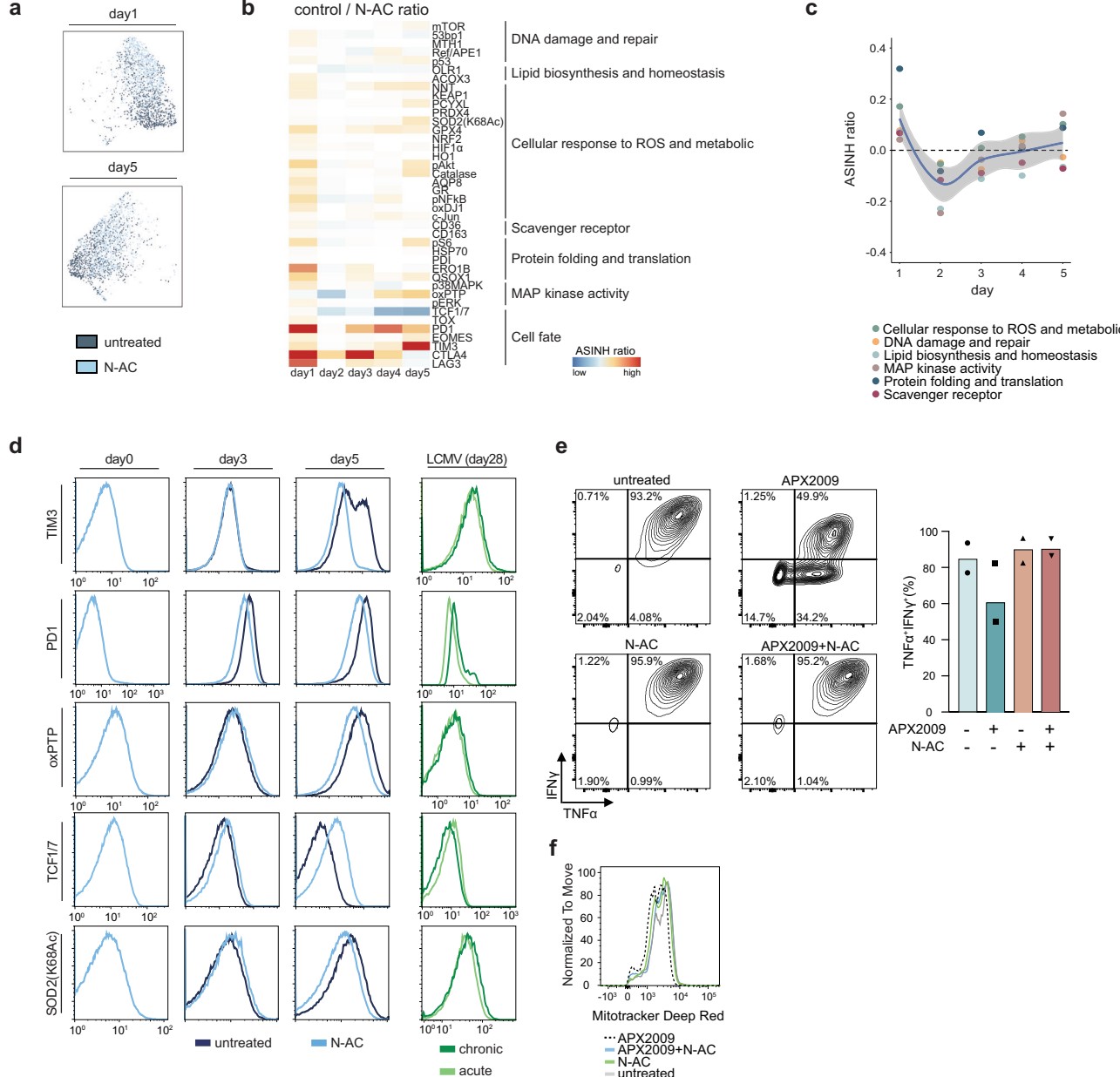

**Fig. 6 | SN-ROP analysis of N-AC-treated CD8⁺ T cells during activation reveal the immediate effects of anti-oxidation treatment in modulating redox pathways. a** UMAP plots of CD8⁺ T cells from OT-1 mice after 1 and 5 days with and without antioxidant N-AC treatment ($n = 2$, data for a representative sample shown) generated using SN-ROP data as input. **b** Heatmap of effect sizes of exhaustion levels in untreated versus N-AC-treated CD8⁺ T cells ($n = 2$). Markers are grouped by GO terms. **c** Loess scatter plot of the ASINH ratio of exhaustion levels in untreated versus N-AC-treated CD8⁺ T cells ($n = 2$). Each point represents a different GO term. The black dashed line represents ASINH ratio = 0. The solid line represents a locally weighted regression (LOESS) fit; the shaded area indicates the 95% confidence interval around the fitted curve. **d** Histogram of average intensities of immune checkpoint inhibitors and SN-ROP markers in CD8⁺ T cells from OT-1 mice cultured

with (dark blue) and without (light blue) N-AC at days 0, 3, and 5 ($n = 2$, data from representative samples shown) and in T cells isolated at day 28 from acute (light green) and chronic (dark green) LCMV models ($n = 3$, data from representative samples shown). **e** Left: Biaxial plots of IFNγ versus TNFα after re-stimulation of T cells from OT-1 mice with PMA and ionomycin without or with APX2009, N-AC, or the combination of APX2009 and N-AC ($n = 2$, biologically independent mice; data for a representative sample shown). Right: Bar plots showing the percentage of TNFα⁺IFNγ⁺ CD8⁺ T cells from individual samples. No statistical comparisons were performed due to limited sample size ($n = 2$). **f** Histogram of MitoTracker Deep Red fluorescence intensity in T cells form OT-1 mice treated with APX2009 (black dashed line), APX2009 and N-AC (blue), N-AC (green), or untreated (gray).

We also investigated whether differential redox states could be detected among hemodialysis patients. We utilized SN-ROP data from 20 hemodialysis patients to train a model that predicts duration from the onset of hemodialysis treatment and assessed its accuracy on the remaining 13 patients. There was a strong correlation between the predicted and actual durations of therapy (Fig. 7e). A significant correlation between the occurrence of sepsis during patient follow-up and

a specific coordinate (coordinate 3) was also identified through multidimensional scaling analysis with an area under receiver operating characteristics curve (AUROC) value of 0.78 (95% CI: 0.6-0.92; Supplementary Fig. 21). Among the redox features analyzed, 43 showed significant correlations with coordinate 3 (Supplementary Table 15). We then applied elastic net logistic regression to identify potential biomarkers of sepsis risk in hemodialysis patients. There were strong

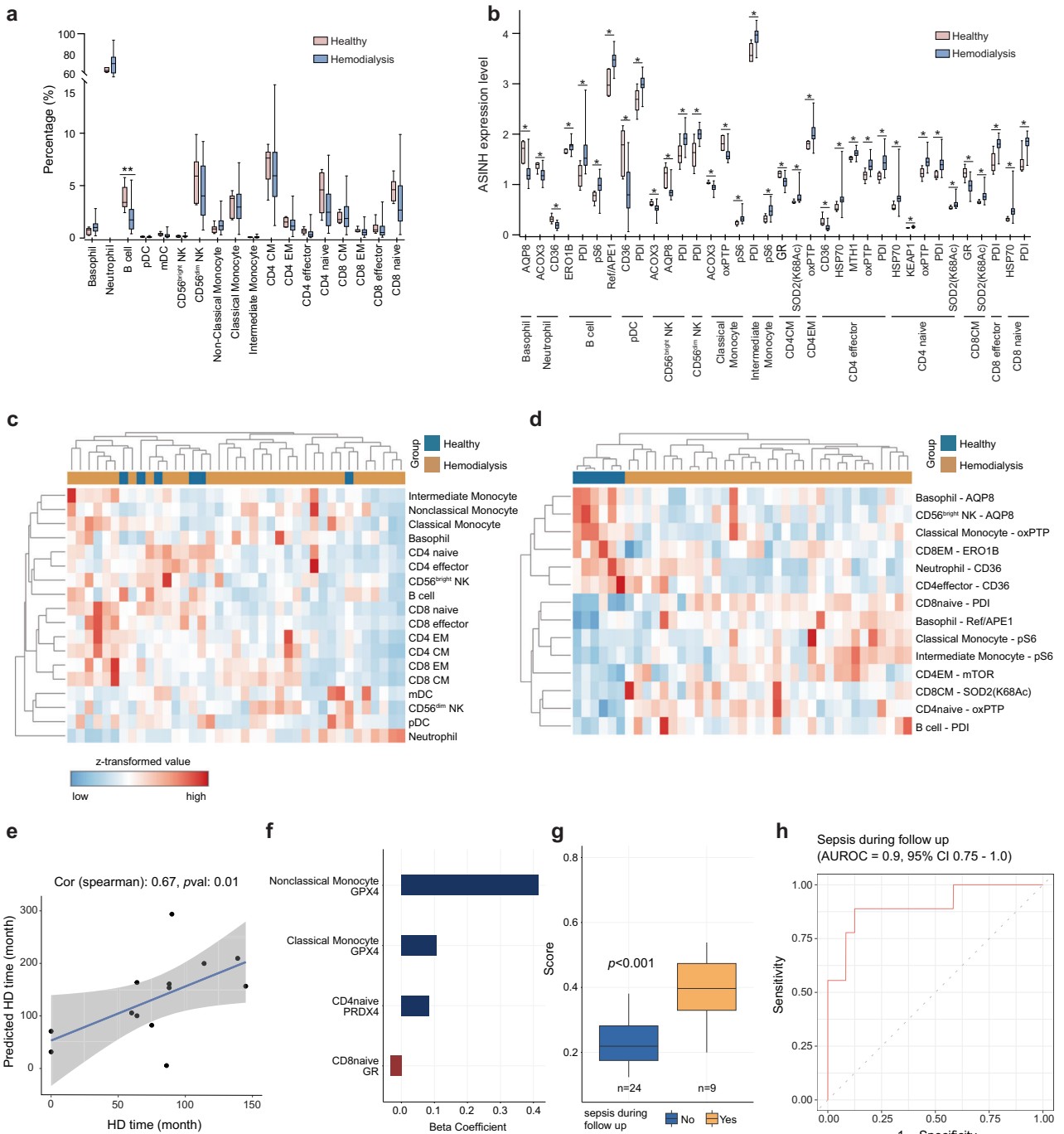

**Fig. 7 | SN-ROP identifies redox features that differ in hemodialysis patients and healthy controls. a** Quantification of the proportion of 18 immune cell types in healthy control individuals ($n = 6$) and hemodialysis patients ($n = 33$). Box plots show the median (center line), the interquartile range (IQR; box limits), and whiskers extending to $1.5 \times$ IQR. Outliers beyond this range are shown as individual points. Statistical significance was assessed using a one-way ANOVA; *$P < 0.05$. **b** ASINH-transformed expression levels of 36 redox-related features across immune subsets in healthy and hemodialysis subjects. Each box represents the distribution of single-cell expression values across samples. Box plots show the median (center line), IQR (box limits), and whiskers extending to $1.5 \times$ IQR; outliers are shown as individual dots. Statistical significance was calculated using two-sided tests with Benjamini–Hochberg correction; *adjusted $P < 0.05$. **c** Clustering diagram generated using only immunophenotyping features for healthy control individuals (blue) and hemodialysis patients (brown). **d** Clustering diagram generated using LASSO focusing on significant ROS features with a false discovery rate (FDR) < 0.2 for healthy control individuals (blue) and hemodialysis patients (brown). **e** Spearman

correlation plot utilizing 18 immune cell types and SN-ROP markers to evaluate the predicted time on hemodialysis (HD) versus the actual time on hemodialysis. Data on two-thirds of the hemodialysis patients (20 patients) were used for training, and data on the remaining one-third (13 patients) constituted the test set. Each dot represents an individual in the test group. The solid line represents the fitted linear regression, and the shaded area indicates the 95% confidence interval. The Spearman correlation coefficient ($\rho$) and two-sided $P$ value are shown in the plot. **f** Beta coefficients of the four ROS features significantly associated with coordinate 3 of the multidimensional scaling plot. **g** Predictive scores derived from elastic net models for hemodialysis patients who developed sepsis ($n = 9$) or did not ($n = 24$) during follow-up. Box plots show the median, interquartile range (IQR), and whiskers extending to $1.5 \times$ IQR. Outliers beyond the whiskers are shown as individual points. Statistical significance was assessed using a two-sided Mann–Whitney U test; *$P < 0.001$. **h** AUROC assessment of the performance of the predictive score in evaluating the risk of sepsis.

associations between sepsis events and the expression levels of GPX4 in classical and alternative monocytes, as well as PRDX4 in naïve CD4[+] T cells and GR in naïve CD8[+] T cells (Fig. 7f, g). The combined AUROC for predicting sepsis events using these four features was 0.9 (95% CI: 0.75-1; Fig. 7h). These results collectively indicate that extended hemodialysis significantly impacts the redox balance in immune cells, potentially compromising infection control. Moreover, our findings underscore the effectiveness of the SN-ROP method in uncovering redox information across various immune subpopulations at the single-cell level.

## Discussion

Within cells, the landscape of redox processes is both highly dynamic and heterogeneous, and these processes are pivotal in the regulation of metabolism, signaling pathways, and the cellular defense against oxidative stress. Previously observed temporal and spatial redox heterogeneity underscores the need for high-dimensional, single-cell approaches to decode the complex redox behavior[34,35]. In this study, we validated SN-ROP, a novel technique that leverages highly multiplexed antibody-based mass cytometry to directly assess redox regulation by quantifying the expression of key markers linked to redox homeostasis. SN-ROP provided an in-depth view of redox states and facilitated a detailed analysis of CD8[+] T cell redox reactions following antigen exposure, under hypoxic conditions, and in the presence of exogenous antioxidants.

Through our SN-ROP analysis, we identified two critical coordinated transition points in redox regulation. As cells shifted from catabolic to anabolic metabolism upon initial antigen stimulation, we detected a marked increase in the activity of mitochondrial superoxide dismutase, a key enzyme in mitigating oxidative stress. This surge was closely followed by redox-related mechanisms involving protein synthesis, folding, and disulfide bond creation, processes that are inherently linked to the generation of ROS that can subsequently trigger enhanced antioxidant responses to preserve redox equilibrium. With continuous antigen stimulation, however, there was a reduction in the efficacy of antioxidant mechanisms and the build-up of oxidized protein byproducts, even when antioxidant protein levels were elevated. The observed impairments in redox regulation within chronically stimulated T cells may be linked to mitochondrial dysfunctions, such as those previously documented in both human exhausted hepatitis B virus-specific CD8[+] T cells and in mice chronically exposed to antigen[36,37]. Mitochondrial ROS production is a crucial signaling element in T cell activation, and an antioxidative glutathione response is indispensable for the metabolic reprogramming post-T cell activation[38,39]. The molecular details of the mechanism by which mitochondria modulate T cell differentiation through metabolic reprogramming and redox signals remain to be elucidated.

The traditional view posits that ROS directly oxidize MAPKs (e.g., ERK[40] and p38[41]), leading to their rapid activation. However, the delay in MAPK activation observed in our study suggests that this process is regulated through indirect mechanisms, such as secondary signaling pathways, post-translational modifications, or redox-sensitive intermediaries, rather than through immediate oxidative modifications. This temporal dissociation indicates that metabolic reprogramming and mitochondrial redox signals may play more critical roles in modulating MAPK activity than previously thought. Our results highlight the complexity of redox signaling in immune cells and suggest that the classical model of MAPK activation should be reassessed. Further work will be needed to elucidate the mechanism that drives the interplay between mitochondrial redox dynamics and T cell signaling pathways.

Using pimonidazole as a surrogate marker for cellular oxygen levels and our SN-ROP method for analysis of the ROS signaling network components revealed a distinct boundary between coordinated transition stages in redox regulation. This suggests that cellular oxygen levels initiate metabolic and redox alterations linked to T cell

exhaustion, aligning with research indicating that HIF1α facilitates the onset of terminal T cell exhaustion[42]. Notably, variations in concentrations of TCF1 and TCF7, which are critical regulators of T cell differentiation[43], were observed during these redox transitions, implying the involvement of redox signaling in adjusting the flexible and reversible functionality of these transcription factors. Future investigations should concentrate on uncovering the molecular processes that facilitate redox signal through TCF1/7. Work should also focus on protein biosynthesis and Ref/APE1, as their co-expression was observed within the same correlated expression cassette as TCF1/7 in our SN-ROP analysis. The connection between Ref/APE1 and proteostasis has been previously reported: Ref/APE1 is induced by stress-mediated activation of the unfolded protein responses[44]. Interestingly, both endoplasmic reticulum stress and the unfolded protein response are implicated in T cell exhaustion[45-47]. Given the sensitivity of Ref/APE1, protein folding, and the unfolded protein response to redox changes[48,49], elucidating the molecular mechanisms that connect these pathways will be important for understanding how T cells adapt to the increased demand for new protein synthesis in response to antigen stimulation and how disruption of these pathways might contribute to T cell exhaustion.

Durable remissions are often achieved with CD19-targeting CAR-T cells, but remission is contingent upon the sustained presence of these cells[50]. We examined the redox states of CAR-T cells over time in patients with hematologic malignancies and discovered a significant correlation between CAR-T cell persistence and their redox activities. As shown in previous transcriptomic analyses[51,52], infusion products displayed early activation patterns, likely stemming from in vitro stimulation during manufacturing. However, distinctive redox profiles emerged in CAR-T cells over time. These profiles categorized patients into groups showing either continual redox activity or a decrease in the redox markers, notably impacting molecules crucial for protein synthesis and folding like ERO1B and PDI. Although the redox-active group showed increased expression of antioxidants, indicators of oxidative stress such as oxPTP and reduced functional performance of SOD2(K68Ac) suggested impaired redox regulation. Notably, our findings revealed a negative correlation between CAR-T cell persistence and the prevalence of redox-active cells, indicating that CAR-T cells must dynamically adapt to environmental stressors and maintain intracellular redox balance for long-term effectiveness. This aligns with recent literature identifying the gene encoding HMOX1 as one of the top genes enriched in persisting CAR-T cells[51] and also suggests that adjusting redox activities shortly after infusion could be a promising approach to enhance CAR-T cell persistence.

End-stage kidney disease impacts more than two million individuals worldwide, with a 5-year survival rate ranging from 41% to 60% for those initiating hemodialysis[53]. Cardiovascular disease is the leading cause of mortality in these patients, trailed closely by sepsis and infection[54]. Using SN-ROP, we identified several redox-specific characteristics in T cells that are notably associated with sepsis during patient follow-up. These findings are consistent with recent research utilizing single-cell RNA sequencing and chromatin accessibility profiling that revealed dysregulation in T cells and monocytes in these patients[55]. Further investigation is warranted to elucidate the mechanisms underlying the dysregulated redox states in T cells from patients undergoing hemodialysis, with a focus on understanding how uremic toxins disrupt T cell metabolism and redox homeostasis, ultimately leading to sepsis[56,57]. Future studies with larger cohorts and extended follow-up periods may uncover additional immuno-redox features correlated with cardiovascular diseases, offering valuable insights into disease pathogenesis and potential therapeutic targets.

In summary, SN-ROP allows precise assessment of redox responses at the single-cell level. As demonstrated here, SN-ROP provides insights into cellular redox dynamics in cell-based systems, mouse models, and human subjects. Its adaptability to diverse cell types

 

makes it applicable across various organ systems. Its versatility will make it applicable to a wide range of biological systems, and it could be employed to aid in the identification of biomarkers and therapeutic targets across diseases. It is important to note that in addition to markers of oxidative stress, SN-ROP also captures broader cellular signaling dynamics that may be influenced by other factors, including metabolic status and environmental conditions. Several limitations of this study should be acknowledged. Redox processes are inherently dynamic and context-dependent, which introduces challenges in capturing the full spectrum of oxidative stress responses. The redox state can be influenced by various factors, including environmental conditions, cell type, and metabolic status, contributing to variability in the observed results. Additionally, although antibody-based detection methods like SN-ROP offer specificity and sensitivity, variability can result from differences in antibody affinity, epitope accessibility, and cross-reactivity. Given these challenges, further validation across various biological contexts will be necessary to confirm the relationship between the redox responses and oxidative stress in specific disease models. Despite these limitations, we believe that SN-ROP will be a valuable platform for future investigations into redox regulation that will significantly impact our understanding of disease mechanisms and therapeutic approaches.

## Methods

### Cells

Raw264.7, SM826, and SH-SY5Y cells were generously provided by Li Chia-Wei, Yang Kai-Chien, Wang Shu-Ping, and Chern Yi-Juang's laboratory at Academia Sinica, Taipei, Taiwan. Raw264.7, SM826, and SH-SY5Y cells were cultured in DMEM (Gibco, 11965092) supplemented with 10% fetal bovine serum (FBS, SH30071.03) and 1% penicillin/streptomycin (Gibco, 15140122). Additionally, SH-SY5Y cells required supplementation with 5 μg/ml blasticidin (Thermo Fisher, A1113903) and 200 μg/ml hygromycin (Thermo Fisher, 10687010). Jurkat cells were cultured in RPMI-1640 medium (Gibco, 11875093) supplemented with 10% FBS and 1% penicillin/streptomycin. HUVEC cells were cultured in Endothelial Cell Growth Medium (PromoCell, C-22010), and HL-1 cells were cultured in Claycomb Medium (Sigma, 51800 C) supplemented with 0.1 mM norepinephrine. All cells were incubated at 37 °C in a 5% $CO_2$ environment. The cells were stimulated with $H_2O_2$ at concentrations of 0, 10, and 100 μM for 0, 0.5, 4, and 48 h. The cells were divided into two groups for analysis. The first group of cells were analyzed using flow cytometry after staining with LIVE/DEAD Fixable Violet Dead Cell Stain (Invitrogen, L34963) diluted 1:1000 in PBS for 30 min at room temperature. The cells were then washed once with PBS and fixed using 1.6% paraformaldehyde (PFA; Electron Microscopy Sciences, 15710) in PBS for 10 min at room temperature. After fixation, the cells were washed and stored in 100 μl aliquots in a 1:10 solution of DMSO (Sigma, D2650) in PBS and stored in −80 °C until flow cytometry analysis. The remaining cells were analyzed by CyTOF according to the staining and acquisition procedures outlined in the Pre-processing and staining of cells for CyTOF section.

### Mice

OT-1 mice (C57BL/6-Tg(TcraTcrb)1100Mjb/J) were purchased from The Jackson Laboratory (Jax 003831). B6 mice (C57BL/6JNarl) were purchased from the Taiwan National Laboratory Animal Center (RMRC 11109). Mice were housed in specific pathogen-free (SPF) facilities at the Academia Sinica SPF Animal Facility, maintained under a 12-h light/dark cycle at an ambient temperature of 22 ± 2 °C and 55 ± 10% humidity. Experimental and control animals were housed in separate cages throughout the study. The protocols for the mouse experiments were approved by the Institutional Animal Care and Use Committee of Academia Sinica (Protocol No. 19-01-1279). All mice used in this study were male and 8–10 weeks old at the time of experiment.

### Human specimens

All patient-related protocols were reviewed and approved by the Institutional Review Boards (IRBs) at the corresponding medical institutes. Samples were collected after obtaining informed consent. Hepatocellular carcinoma patients and healthy donors for the lineage analyses were recruited at the National Taiwan University Hospital, Taipei, Taiwan (IRB No. 201912040RINA). Hemodialysis patients were recruited at Kaohsiung Medical University Hospital (IRB No. KMUHIRB-E(I)−20200109). CAR-T patient samples were collected at the National Taiwan University Cancer Center as part of two interventional clinical trials registered at ClinicalTrials.gov (NCT04943016 and NCT03624686). The studies were approved by the Institutional Review Board of the National Taiwan University Hospital (IRB No. 202103150MIPC and 201711021RIND), and written informed consent was obtained from all participants.

### Metal conjugation of antibodies for CyTOF analysis

Primary conjugates of mass cytometry antibodies were generated using the Maxpar X8 Antibody Labeling Kit (Fluidigm, 201300) following the manufacturer's instructions. After labeling, the antibodies were diluted to a final stock concentration of 0.5 mg/ml in Candor PBS Antibody Stabilization Solution (Candor Bioscience GmbH, 131050) containing 0.02% $NaN_3$ (Sigma, S2002) and stored at 4 °C for long-term use. Each antibody clone and lot was titrated to determine optimal staining concentrations using appropriate positive and negative controls.

### Pre-processing and staining of cells for CyTOF

Frozen cells were thawed. For pre-processing, cells from each study were first washed once with 1 ml of serum-free RPMI medium (Gibco, 11875093). Then, the cells were stained with cisplatin (Sigma, P4394) at a final concentration of 25 μM for 1 min at room temperature to label dead cells. The stain was quenched by adding 1 ml of RPMI medium with 10% FBS (Cytiva, SH30071.03) for 3 min. The cells were then fixed in 1.6% PFA (Electron Microscopy Sciences, 15710) in PBS (Corning, 46-013-CM) at room temperature for 10 min. Following fixation, the cells were washed three times with CSM (PBS with 0.5% protease-free bovine serum albumin [Sigma, A3059] and 0.02% $NaN_3$ [Sigma, S2002]). Finally, the cells were stored by freezing them in a 1:10 solution of DMSO (Sigma, D2650) in PBS in 100-μl aliquots.

Palladium barcoding was performed using the Cell-ID 20-Plex Pd Barcoding Kit (Fluidigm, Fluidigm, 201060) following the manufacturer's protocol[58]. Briefly, individual samples were incubated with unique palladium-based barcodes in barcode perm buffer for 30 min at room temperature, then washed twice with CSM. Samples were combined for processing. The cell surface antibody cocktail mix in CSM was filtered through a pre-wetted 0.1-μm spin-column (Millipore, UFC30VV00) to remove antibody aggregates and was added to the combined samples. After incubating at room temperature for 1 h, the cells were washed with CSM. For intracellular staining, the cells were permeabilized with ice-cold methanol (Fisher Scientific, A454-4), washed to remove residual methanol, and then incubated with the intracellular antibody cocktail mix at room temperature for 1 h. After another wash with CSM, the cells were suspended and stained with Cell-ID Intercalator-Ir ($^{191}$Ir and $^{193}$Ir; Fluidigm, 201192B) at final concentrations of 125 nM in 500 μL 4% fresh PFA diluted in PBS overnight at 4 °C to stain DNA. The cells were washed with CSM and filtered, resuspended in a solution containing EQ Four Element Calibration Beads (Fluidigm, 201078), and analyzed using a CyTOF2 mass cytometer (Fluidigm).

### Antibody screening and validation

Samples of cells collected at indicated times points after stimulation were thawed, washed once with PBS (Corning, 46-013-CM), and stained using a live-cell amine-reactive fluorescent labeling protocol[59]. Briefly,

the cells were stained using amine-reactive fluorescent dyes, specifically Pacific Orange (Thermo Fisher, P30016) and DyLight 350 (Thermo Fisher, 46426), at concentrations of 0, 0.1, or 1 μg/ml. Some samples were labeled with Alexa Fluor 546 (Invitrogen, A-20102) at concentrations of 0, 0.1, or 1 μg/ml and Alexa Fluor 700 (Invitrogen, A-20010) at concentrations of 0, 0.01, 0.07, or 1 μg/ml. The antibodies were detected using Alexa Fluor 488-conjugated secondary antibodies. The data were acquired using an LSRII HTS cytometer (BD Biosciences) and analyzed using FlowJo software (FlowJo, LLC, https://www.flowjo.com/). Analysis was performed based on fluorescence minus one or biological comparison controls for accurate gating and interpretation of the results. For the selection of antibodies to construct the SN-ROP panel, the ASINH- transformed mean fluorescence intensity at 0 h−averaged across three independent replicates−served as the baseline for each antibody. Antibodies that showed more than a 10% deviation from this baseline under any condition (72 out of 103) were selected for further analysis. A correlation matrix was constructed to examine the relationships among these 72 antibodies, and they were grouped into seven modules based on their co-regulation patterns (Supplementary Fig. 1b). Within each module, antibodies were ranked according to a weighted average score reflecting their relative contribution. This weighted score determined how many were selected, ensuring that the most responsive antibodies were included while maintaining an appropriate balance across the modules. All antibodies used in the screening were stained at a 1:100 dilution.

### Isolation and stimulation of CD8+ T cells from OT-1 mice

Single-cell suspensions were prepared from the spleens of OT-I mice (Jackson, 003831). The cells were cultured at a concentration of $1 \times 10^6$ cells per ml in RPMI-1640 medium (Gibco, 11875093) supplemented with 10% FBS (Cytiva, SH30071.03), 2 mM L-glutamine (Thermo Fisher, A2916801), 50 μM β-mercaptoethanol (Sigma, SI-M3148), 20 ng/ml IL-2 (PeproTech, 212-12), 5 ng/ml IL-7 (PeproTech, 217-17), and 5 ng/ml IL-15 (PeproTech, 210-15) in the presence of 10 ng/ml SIINFEKL peptide (InvivoGen, OVA 257−264) for 48 h. For short-term treatment, the SIINFEKL peptide was not added after the initial 48-h stimulation, and medium was replaced with RPMI-1640 medium containing 10% FBS, 2 mM L-glutamine, and 5 μM β-mercaptoethanol, supplemented with 10 ng/ml IL-2, 5 ng/ml IL-7, and 5 ng/ml IL-15 at a concentration of $1 \times 10^6$ T cells per ml. For longer treatment, 1 μM SIINFEKL was added to the medium. The T cells were passaged into fresh co-cultures every 48 h until the end of the experiment. N-AC (Sigma, A7250) was dissolved in water and used at a concentration of 10 mM. APX2009 (Sigma, SML1887) was dissolved in DMSO (Sigma, D2650) and used at a concentration of 1 μM. For experiments under normoxic or hypoxic conditions, cells were cultured in normoxic (20% oxygen) or hypoxic (1.5% oxygen) chambers at 37 °C. Cells were cultured with 200 mM pimonidazole (Hypoxyprobe, Inc., HP10-1000mg) for a duration of 4 h and then stained with a biotin-conjugated antibody, which is included in the Hypoxyprobe Kit, at a dilution of 1:500 in PBS (Corning, 46-013-CM) for 10 min. Subsequently, the cells were washed with PBS and then subjected to cisplatin staining, fixation, and storage as outlined in the Pre-processing and staining of cells for CyTOF section.

### MC38 tumor-infiltrating lymphocyte dissociation

B6 mice were injected subcutaneously with $1 \times 10^6$ MC38 colon adenocarcinoma cells (a gift from Dr. Hu Che-Ming, Academia Sinica). Tumor growth was monitored every 2−3 days using digital calipers, and tumor volume was calculated using the formula: $(\text{length} \times \text{width}^2)/2$. Mice were euthanized by $CO_2$ inhalation followed by cervical dislocation on day 7 or day 14 after tumor injection or earlier if tumors reached 1.5 cm in diameter or if signs of ulceration, impaired mobility, or severe distress were observed. All procedures were conducted in accordance with the guidelines of the Institutional Animal Care and Use Committee (IACUC) of Academia Sinica (Protocol No. 19-01-1279),

and no tumors exceeded the permitted size limits. Tumor tissues were collected and dissociated in RPMI medium (Gibco, 11875093) supplemented with 10% FBS (Cytiva, SH30071.03), collagenase IV (Worthington Biochemical, LS004188; 1 mg/ml), and DNase I (Roche, 11284932001; 0.1 mg/ml) to dissociate the tumor cells. The dissociated cells were analyzed by CyTOF according to the procedures described in the Pre-processing and staining of cells for CyTOF section.

### Human sample preparation

For the healthy donor lineage analysis, human whole blood samples from healthy donors were collected using an EDTA anticoagulant (BD, PS8-7525). The collected blood samples were then diluted 1:10 in 1 × ACK lysing buffer (Gibco, A10492-01) to lyse red blood cells. After 5 min, cells were washed with PBS (Corning, 46-013-CM). For the hemodialysis patient study, whole blood samples were collected from 6 healthy control donors and 33 patients. The samples were frozen immediately. Red blood cells were lysed as per the protocol from CYTODELICS.

For the CAR-T study, peripheral blood mononuclear cells (PBMCs) from CAR-T patients were isolated using density gradient centrifugation with Ficoll-Paque PLUS (GE Healthcare, 17-1440-02). The isolated PBMCs were then frozen and stored. Once all the time-course samples were collected, the frozen cells were thawed and stained with a 1:50 human CD19 CAR Detection Reagent (Miltenyi Biotec, 130-115-965) in PBS for 30 min on ice to label the CAR-T cells. After staining, the cells were washed and fixed in 1.6% PFA (Electron Microscopy Sciences, 15710) in PBS at room temperature for 10 min, then washed three times with CSM. Cells were then pre-processed, barcoded, and stained with the CyTOF ROS panel according to the procedures detailed in the Pre-processing and staining of cells for CyTOF section.

### Immune cell isolation from the liver tissue

Liver tissues from patients with hepatocellular carcinoma were dissected in the operating room. The normal, border, and tumor core regions were visually confirmed and separated. Fresh tissues were dissociated into small pieces in a Petri dish containing a small amount of MACS buffer (PBS [Corning, 46-013-CM], 2% FBS [Cytiva, SH30071.03], 2 mM EDTA [BD, PS8-7525]). The tissues were transferred onto a 70-μm cell strainer (Falcon) affixed on top of a 50-ml centrifuge tube. A 5-ml syringe was used to press and mince the tissue. MACS buffer was added to facilitate the flow-through of the dissociated cells into the centrifuge tube. The dissociated cells were centrifuged at 500 rpm for 2 min to pellet the hepatocytes. To collect the immune cells, the supernatant was aspirated and transferred to a new centrifuge tube. The immune cell-containing solution was centrifuged again at 1500 rpm for 10 min. The supernatant was discarded, and the cell pellet was washed twice with PBS. The cells were then prepared for CyTOF.

### LCMV infections

For acute LCMV infections, C57BL/6 (B6) mice were injected intraperitoneally with $2 \times 10^5$ plaque-forming units (PFUs) of the LCMV Armstrong strain 53b. For chronic infections, B6 mice received an intravenous injection of $2 \times 10^6$ PFUs of the LCMV Clone 13 strain. Viral titers were measured in the spleen, kidneys, and blood. Organ suspensions were prepared from these tissues and used to infect Vero cells (ATCC CCL-81), with viral loads quantified through an LCMV focus-forming assay. This infection model is commonly used to induce acute or chronic viral infection and to study T cell activation and exhaustion dynamics in vivo[60].

### MitoTracker staining

To assess mitochondrial membrane potential and mass, cells were incubated with 10 nM MitoTracker Deep Red (Thermo Fisher Scientific, M22425) in RPMI medium (Gibco, 11875093) supplemented with

2% FBS (Cytiva, SH30071.03). The staining was performed for 15 min at 37 °C. After incubation, the cells were washed and analyzed using a LSRII flow cytometer (BD Biosciences) with BD FACSDiva software (v8.0.1). Data were analyzed using FlowJo software (FlowJo, LLC).

## Trajectory analysis of ROS remodeling

In each condition, a random subsample of 2000 cells were selected from each day. The subsampling and subsequent analysis were performed using UMAP from Cytobank (https://www.cytobank.org/), a platform for high-dimensional cytometry data analysis. Pseudotime analysis was conducted using the SCORPIUS algorithms[61]. The resulting pseudotime values were scaled from 0 to 1. Resting cells (day 0) and cells from day 5 were included in the trajectory calculation to identify starting and ending points, but the focus of interest was primarily on the intermediate time points (pseudotime 0.1–0.9).

For visualization purposes, the pseudotime heatmap data were 99th percentile normalized and were plotted using Morpheus (https://software.broadinstitute.org/morpheus/). The derivative pseudotime heatmap was then calculated by determining the slope from the pseudotime heatmap. The pseudotime line chart was generated using the data from the pseudotime heatmap, smoothed using a window size of 1000. Lastly, the pseudotime pathway heatmap was generated using the data from the pseudotime heatmap, with calculations performed for different pathways and smoothing applied using a window size of 100.

## Gene ontology enrichment analyses

Annotations were obtained from UniProt (https://www.uniprot.org/), which provides information on GO terms, biological processes, and pathways. The SN-ROP data were categorized and assigned to relevant functional modules or pathways based on the following annotations: DNA damage and repair (53bp1, MTH1, p53, Ref/APE1, and mTOR); lipid biosynthesis and homeostasis (ACOX3 and OLR1); cellular response to ROS and metabolic (AQP8, Catalase, GPX4, GR, HO1, KEAP1, NNT, NRF2, PCYXL, PRDX4, SOD2(K68Ac), oxDJ1, HIF1α, pNFκB, c-Jun, and pAkt); scavenger receptor (CD163 and CD36); protein folding and translation (ERO1B, HSP70, PDI, QSOX1, and pS6); and MAP kinase activity (oxPTP, pERK, and p38MAPK).

## Machine learning methods

The machine learning models for the blood cell lineage prediction were developed using Python 3.9.5 and the following libraries: scikit-learn 1.2.1, xgboost 1.6.2, and Catboost. Several supervised machine learning algorithms were tested for classifying different immune cell types using ROS markers as features. The algorithms included Random Forest, 3-Layer Feed-forward Neural Network, Quadratic Discriminant Analysis Classifier, AdaBoost, Naive Bayes Classifier, Decision Tree, K-Nearest Neighbors, XGB Classifier, and CatBoost Classifier. The most accurate algorithm, Catboost, was selected as the main algorithm for training the model (Supplementary Table 16). Specifically, we employed 10-fold cross-validation with 10 runs, using different random states for each run. This allowed us to evaluate the robustness and consistency of the model across multiple random states. The dataset was divided into two groups: 80% for training and 20% for testing. To test the generalizability of the model, interindividual variability of ROS markers was assumed to be negligible. In the case of healthy donor samples, data from eight donors were used to train the model, and data from the remaining two donors were used for testing. For the in vivo versus in vitro timeline projection study, 80% of the cells were used for training, and the remaining 20% were used for testing.

To build a prediction model for the hemodialysis duration using the ROS features, the dataset was split into a training set and a testing set in a 2:1 ratio. The ROS features were standardized through Z transformation with means and standard deviations of the respective variables. We used an elastic-net model to perform feature selection.

We optimized the parameter by conducting 10-fold cross-validation on the training set, selecting the one with the smallest mean squared error. To evaluate the performance of the prediction model, we calculated the Spearman correlation coefficient between the predicted dialysis time (in months) and the actual dialysis time on the testing set. We aimed to determine if the ROS features more effectively differentiate between dialysis patients and healthy controls than immunophenotyping features. To achieve this, we visualized the classification outcomes using heatmaps and average-link hierarchical clustering based on the distance matrices of pairwise squared ranking differences between features. Lasso regression was utilized to select relevant ROS/immunophenotyping features for classification and visualization. All analyses were conducted using R 4.3.1. The multidimensional scaling analysis was performed using the cmdscale() function in R on the distance matrix derived from the selected features, followed by principal coordinate plotting to assess group-wise differences[62].

## DREVI plot generation

DREVI was used to visualize the stochastic function that represents the influence of hypoxic cells (identified by pimonidazole labeling) on the SN-ROP data. DREVI was used to plot the distribution of SN-ROP values for each hypoxic cell condition. The DREVI plot was generated by analyzing individual FCS files using the software tools available at (https://www.cytobank.org/). During the analysis, doublets, debris, and dead cells were excluded based on criteria such as cell length, DNA content, and cisplatin staining. All surface markers were taken into account for gating out purified CD8+ T cells. The resulting data were exported for DREVI plot generation using the software package accessible at (www.c2b2.columbia.edu/danapeerlab/html/dremi.html).

## Replication and ASINH ratio calculation

The replication heatmap visualizes the ASINH-transformed results of replicated data using the pheatmap package in R language, incorporating unsupervised hierarchical clustering. The ASINH ratio was calculated using the ASINH-transformed data. First, the average of each condition and time point for the replicated pairs was computed. Then, the control was subtracted from the treatment group for each pair (for example, data for hypoxic conditions at day 0 minus data for normoxic conditions at day 0). This process was repeated for all pairs. The ASINH ratio analysis was performed using the pheatmap package in R.

## Measurement of cytokine production

T cells were stimulated with 50 ng/ml phorbol 12-myristate 13-acetate (PMA; Sigma, P8139) and 500 ng/ml ionomycin (Sigma, I0634), together with 1:1500 dilutions of brefeldin A (Invitrogen, 00-4506-51) and monensin (Invitrogen, 00-4505-51) in complete medium. After 4 h of incubation, cells were washed and fixed in 1.6% paraformaldehyde (Electron Microscopy Sciences, 15710) in PBS for 10 min at room temperature. Following fixation, cells were washed and permeabilized using the eBioscience™ Intracellular Fixation & Permeabilization Buffer Set (Thermo Fisher, 00-5523-00) for 15 min. Cells were subsequently washed and stained with surface and intracellular antibodies, including anti-CD3 (BioLegend, 100249; clone 17A2; BV785; 1:200 dilution), anti-CD8α (BioLegend, 100712; clone 53-6.7; APC; 1:200 dilution), anti-IFNγ (BioLegend, 505808; clone XMG1.2; PE; 1:25 dilution), and anti-TNFα (BioLegend, 506313; clone MP6-XT22; Alexa Fluor 488; 1:50 dilution), for 40 min at room temperature. Following staining, cells were washed twice with PBS and analyzed using an LSRII flow cytometer (BD Biosciences). Mean fluorescence intensity (MFI) values were quantified using FlowJo software (v10.0; FlowJo, LLC).

## Validation experiments

Mass spectrometry data were obtained from a publicly available proteomic resource (http://www.immprot.org/), and RNA-seq data for the

Jurkat cell line were obtained from a public database (https://www.tandfonline.com). Log10 values were calculated for each SN-ROP marker in 18 immune cell subsets. Mass cytometry results were analyzed using Cytobank and are depicted as ASINH-transformed expressions within the 18 immune cell subsets. Pearson correlation coefficients were calculated. RNA-seq data were analyzed to calculate the fold change between $H_2O_2$-treated cells (48 h, 100 μM) and untreated cells (0 h, 0 μM). CytoScore is the average localization of SN-ROP components HSP70, KEAP1, oxPTP, MTH1, and GR in the cytoplasm, whereas MitoScore is the average expression of mitochondria-related antibodies oxDJ1, SOD2(K68Ac), NNT, and GPX4.

## Visualization and statistical analyses

Line-and-box plots were prepared using GraphPad Prism. Fitting of generalized linear models and visualization were performed using R packages including SCORPIUS, ggplot2, and pheatmap. $P < 0.05$ was taken as statistically significant. Schematic representations were created using BioRender.com under an Academia Sinica - Life Science Library license with publishing rights. The final figures were prepared using Adobe Illustrator.

## Reporting summary

Further information on research design is available in the Nature Portfolio Reporting Summary linked to this article.

## Data availability

Mass cytometry data sets for SN-ROP analysis of healthy human, CAR-T, hepatocellular carcinoma, and hemodialysis patient whole blood populations, OT-I T cells, and MC38 mouse tissues are publicly available at Zenodo via the https://doi.org/10.5281/zenodo.11541294. All data are included in the Supplementary Information or available from the authors upon reasonable request, including unique reagents used in this Article. The raw numbers for all charts and graphs are available in the Source Data file whenever possible. Source data are provided with this paper.

## Code availability

The machine learning-based ROS analysis code used in this study is publicly available on GitHub at [https://github.com/SYChenLab/ROS]. To ensure reproducibility and long-term accessibility, a versioned snapshot has also been published on Code Ocean with [https://doi.org/10.24433/CO.4463135.v1]. The capsule includes the full execution environment and instructions for replicating the key analyses in this study. Additionally, code specific to the hemodialysis sepsis prediction model has been deposited on Zenodo (https://doi.org/10.5281/zenodo.11541294) and is accompanied by comprehensive documentation. All code is available under a permissive open-source license.

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

## Acknowledgements

We thank the medical staff who participated in patient treatment, the doctors who collected specimens, and the patients who provided them. The authors are grateful for technical support from the CyTOF center of the Core Research Laboratory, College of Medicine, National Cheng Kung University and Common Mass Spectrometry Facilities for Proteomics and Protein Modification Analysis located at the Genomics Research Center, Academia Sinica (AS-CFII-108107). We thank staff at the Academia Sinica Core Facility and Innovative Instrument Project (AS-CFII-111-212) for cell sorting service and analysis of flow cytometry. We thank the Academia Sinica SPF Animal Facility for providing animal support. The core facility is funded by Academia Sinica Core Facility and Innovative Instrument Project (AS-CFII-111-204). We thank Ya-Ping Lin and Jung-Lee Lin for mass cytometry technical contributions. BioRender.com was used to create the schematic summary depicting our proposed model, under an Academia Sinica - Life Science Library license with publishing rights. This study was supported by National Taiwan University Hospital (NTUH 111-T0010, NTUH 113-L3004), National Taiwan University Cancer Center (NTUCCS-114-04, NTUCCS-113-06, and NTUCCS-111-01), Academia Sinica (Career Development Award [S-YC], AS-CDA-110-L09, AS-GC-110-05, and AS-KPQ-110-EIMD, AS-GCS-111-L03, AS-GC-110-MD04, and VTA111-V3-1-2), Kaohsiung Medical University Hospital, Taiwan (KMUH-DK(B)110003-1, KMUH110-0R19, KMUH110-0M13, KMUH111-1M09, KMUH111-1R14), Kaohsiung Medical University (NHRIKMU-111-I001-3, NHRIKMU-111-I003, NHRIKMU-111-I003-1), the Ministry of Science and Technology (MOST 109-2314-B-037-088, MOST 109-2314-B-037-102-MY2, MOST 109-2326-B-002 –012 -MY3, MOST 112-2320-B-001-014-, MOST 112-2320-B-001-029-, MOST 112-2314-B-002-091-MY3, MOST 111-2320-B-001-019-, MOST 110-2320-B-001-024-, and 112-2314-B-002-091-MY3), and the Swiss Science National Foundation Consolidator Grant (TMCG-3_213736), the Cancer Research Institute (Lloyd J. Old STAR award), and the Helmut Horten Stiftung.

## Author contributions

Y.C.W. and S.Y.C. designed and performed experiments, analyzed and interpreted data, created figures, and wrote the manuscript. Y.M.C. and Y.T.H. conceived and computationally implemented the conceptual framework and interpreted data. M.H.Y., Y.F.W., Y.M.C., and J.Y.S. coordinated computational analysis. J.L.T., P.S.W., and T.H.S. obtained clinical and pathological information. W.C.T., H.I.S., P.L.T., W.M.H., and M.C.K. coordinated patient sample collection. L.W. performed mouse

sample collection. Y.S.T. wrote the manuscript. G.W., S.L.L., C.W.L., T.M.K., K.C.Y., Y.J.Chang., Y.J.Chern., P.C.H., C.T.W., Y.L.C., T.C.C., and T.C.M. provided technical support and funding acquisition. Authors marked with † contributed equally to this work: P.H.W., W.C.T., Y.F.W., M.H.Y., T.H.S., and J.Y.S. All authors revised the manuscript and approved its final version.

## Competing interests

The authors declare no competing interests.

## Additional information

[1]Program in Molecular Medicine, National Yang Ming Chiao Tung University, Taipei, Taiwan. [2]Institute of Biomedical Sciences, Academia Sinica, Taipei, Taiwan. [3]Division of Nephrology, Department of Internal Medicine, Kaohsiung Medical University Hospital, Kaohsiung Medical University, Kaohsiung, Taiwan. [4]Faculty of Medicine, College of Medicine, Kaohsiung Medical University, Kaohsiung, Taiwan. [5]Biomedical Artificial Intelligence Academy, Kaohsiung Medical University, Kaohsiung, Taiwan. [6]Tai-Chen Stem Cell Therapy Center, National Taiwan University, Taipei, Taiwan. [7]Retain Biotech, New Taipei City, Taiwan. [8]Division of Gastroenterology and Hepatology, Department of Internal Medicine, National Taiwan University Hospital, Taipei, Taiwan. [9]Institute of Statistical Science, Academia Sinica, Taipei, Taiwan. [10]Department of Pathology, Institute of Stem Cell Biology and Regenerative Medicine, Stanford University School of Medicine, Stanford, CA 94305, USA. [11]Department of Oncology, University of Lausanne, Lausanne, Switzerland. [12]Ludwig Institute for Cancer Research, University of Lausanne, Epalinges, Switzerland. [13]Department of Pathology, National Taiwan University Cancer Center, National Taiwan University Hospital and National Taiwan University College of Medicine, Taipei, Taiwan. [14]Department of Otolaryngology, National Taiwan University Hospital and National Taiwan University College of Medicine, Taipei, Taiwan. [15]Institute of Biological Chemistry, Academia Sinica, Taipei, Taiwan. [16]Institute of Biochemical Sciences, National Taiwan University, Taipei, Taiwan. [17]Department of Biological Sciences and Technology, College of Engineering Bioscience, National Yang Ming Chiao Tung University, Hsinchu, Taiwan. [18]Division of Cardiology, Department of Internal Medicine, National Taiwan University Hospital, Taipei, Taiwan. [19]Department and Graduate Institute of Pharmacology, National Taiwan University, Taipei, Taiwan. [20]Department of Hematological Oncology, National Taiwan University Cancer Center, National Taiwan University Hospital, Taipei, Taiwan. [21]These authors contributed equally: Ping-Hsun Wu, Wen-Chieh Ting, Yi-Fu Wang, Ming-Han Yang, Tung-Hung Su, Jia-Ying Su. ✉e-mail: sychen@ibms.sinica.edu.tw

