## [Transparent Peer Review file · Nature Communications]

Single-cell signaling network profiling during redox stress reveals dynamic redox regulation in immune cells

Corresponding Author: Dr Shih-Yu Chen

Version 0:

Reviewer comments:

Reviewer #1

(Remarks to the Author)

In this manuscript, the authors investigate reactive oxygen species at the single cell level that they call a single-cell ROS regulome profiling method. They initially carry out experiments in individual cell lines to develop their targets. Then in blood samples from healthy donors, finally in patients with medical conditions: kidney dialysis, hemophilia, blood cancers. I found the paper fascinating and readable, but my knowledge of biology is not at a level where I can make meaningful comments on these studies.

Here are comments that I think the authors should take into account in a revised submission.

Methods. In the Methods section there is no information provided about antibody labeling for CyTOF measurements. In the Introductions and Results section, the authors write "We first conducted a thorough screening of more than 100 antibodies targeting redox-related proteins." "From this vast screening, 25 antibodies were chosen for their strong correlations with standard ROS measures." Presumably these are the antibodies stained with Maxpar reagents (Fluidigm/Standard BioTools) for their CyTOF experiments. Even more curious, the term Maxpar does not appear anywhere in the manuscript.

Tables at the end of the manuscript identify which metals are on each antibody. Several of the Tables have no title. There is a Supplementary Table 3 Supplementary Table 2 or 1.

Reviewer #2

(Remarks to the Author)

This manuscript by Wang et al. develops a new single cell assay (scROP) that attempts to bring redox biology to the single cell arena and applies the assay to CAR-T therapy in patients. The potential is highly significant, since ROS and redox-focused assays are technically challenging they are rarely comprehensive nor amenable to single cell analysis. The approach taken by the authors is ambitious, and the assay itself is a significant advance in the field. Furthermore, their application to CAR-T patients is also ambitious, and has high clinical relevance, and clearly interesting patterns emerge from the scROP results demonstrating its utility. However, despite these strengths, there are several major concerns detailed below.

One major concern is related to accuracy of the claims, as this assay is not comprehensive, nor is there is their clear evidence that the assay is even specific for redox biology. scROP is more of a generalized signaling assay that incorporates some minor input from ROS and redox biology, but most of these markers are also regulated by non-redox stimuli and the ROS-specific markers do not appear to be terribly informative in the data presented. Thus, the scROP assay is useful, but the relevance of redox biology per se isn't clear and the specificity for redox processes is quite unlikely. A second major concern is that nearly all the claims are associative or correlative, with very limited functional studies aside from inhibition of REF-1 that was driven by knowledge of redox biology as opposed to being revealed by the scROP assay results themselves. Even in the abstract, the phrases used are vague, including "established a connection between", "linked", and "significant correlation".

Main concerns:

1. The authors routinely claim they are 'comprehensively' measuring the 'ROS regulome', for example line 185 states "To

investigate redox states comprehensively, we aimed to quantify the abundance of ROS transporters, pivotal ROS-generating and ROS-scavenging enzymes and their regulatory modifications (e.g., phosphorylation), products of prolonged oxidative stress (e.g., sulfonic oxidation modification of proteins), and the transcription factors and signaling molecules that drive specific redox programs, collectively termed the cellular ROS regulome” Yet, they aren’t measuring any NOX, only 1 of 6 PRDXs, and only a single Aquaporin. Thus the data provided isn’t comprehensive, and wouldn’t be sufficiently complete to understand the architecture of ROS production. Moreover, they claim on line 167 that they’ve developed a “panel that allows comprehensive exploration of how oxidative stress is generated and neutralized as well as impacted mechanisms” which is vastly overstated based on the limited markers assayed. This assay is nowhere near capable of ‘omics’, and claiming that this assay can measure anything close to a regulome need to be dramatically reduced.

2. Relatedly, nearly all of the antibodies chosen are not redox specific. Phospho-ERK and phospho-AKT, and most protein expression, are also affected by many non-redox perturbations. It is therefore unclear much influence redox biology has on the observed changes as opposed to non-redox-dependent change. For example, how much of the time-dependent changes observed by scROP in Figure SA really depend on redox biology? It seems to be low since no redox-specific marker (e.g. oxDJ1, oxPTP) is highlight in the next panel, Figure 2B. Insight into how much redox biology influences this could be gleaned comparing scROP assay results upon addition of an antioxidant, but otherwise the results do not demonstrate that the scROP assay is highly specific for redox signaling / biology. This is a major concern since the assay is positioned to measure redox biology, even the redox ‘regulome’, and re-positioning away from a narrow focus on redox biology is needed. For example, many of the observed changes in Fig. 2C happen before any increase in oxidization of DJ1 or oxPTP, which would presumably measure the onset of ROS production, clearly indicating non-redox-dependent change are detected by this assay.

Similarly, the ‘active’ versus ‘basal’ types of CAR-positive T cells (Fig. 3B), are really driven by 4 markers, protein expression of NNT, GPX4, HSP70 and phosphorylation of p38MAPK. This is hardly a change in the redox regulome, and it’s hard to claim that these markers are even redox-specific. In short, the scROP assay is useful, and may include some aspects of redox biology, but its specificity to redox biology is very inconclusive.

3. The authors state that they ‘validated’ their assay in Figure 1, claiming that their MitoScore (an expression measure) matches levels of bona fide mitochondrial ROS levels using mitoSOX. However, the validation criteria are not stated, and a simple experiment generating mitochondria ROS and observing an increase in their mitoROS proxy (MitoScore) would be excellent validation. However, this isn’t shown and it’s unlikely that mitochondrial protein expression correlates with mitochondrial ROS. Additional validation is needed to make claims about being able to assess mitochondrial ROS using this assay.

4. The functional utility of the scROP assay isn’t established. While the authors use a REF-1 inhibitor in figure 5, this choice isn’t driven by scROP assay results. Nearly all the claims are associative or correlative, with very limited functional studies aside from inhibition of REF-1 that was driven more by knowledge of redox biology as opposed to being driven from scROP assay results. Even in the abstract, the phrases used of “established a connection between”, “linked”, and “significant correlation”. The descriptive nature of these studies limits their significance.

Minor points:

- The authors claim that Fig. 1F shows that each type of normal immune population has a “discovered that each cell type has a unique redox pattern”, but B and T cells look quite similar, as do basophils and NK cells. More rigorous analysis is needed to make this claim, especially since Fig 1g shows that the scROP pattern of many of these immune cells overlap.
- Details of the machine learning model are insufficient. The authors state that “The most accurate algorithm, Catboost, was selected as the main algorithm for training the model.”, but how was this accuracy assessed and was this based on training data, testing data, etc? Also, on what basis did the authors decide that “interindividual variability of ROS markers was assumed to be negligible”? Was the testing performed for only a single random state and random assignment of samples to Test? If so, that is insufficient. Lastly, it is unclear how the authors went from a correlation (“Spearman correlation coefficient between the predicted dialysis time (in months) and the actual dialysis time”) to the accuracy and ROS plot shown in Fig. 1H.
- Given the importance of the MitoScore to validation of this assay, it would be useful to see how measure that changes over pseudotime in the relevant figures and machine learning-based classifications.
- The result (Fig. 2G) that protein folding and translation happens well before activation of most pathways and “Cellular response to ROS” along with the notable delay in MAP kinase activity is very interesting and a unique insight that this assay could provide. This should be discussed, especially since it seems like MAPK-dependent activation by direct oxidation (the canonical model) doesn’t appear to be occurring.
- The authors state that Fig. 2F “identified four distinct groups of features with analogous correlation patterns”, but this isn’t clearly indicated.

Reviewer #3

(Remarks to the Author)

This manuscript describes an enormous undertaking in which the authors have used the newish and powerful technique of mass cytometry to examine proteomic changes in single cells. While there are other methods that can provide similar information for a population of a particular cell type, a major advantage of this technique is that it can be applied to heterogeneous cell populations. The technique and the bioinformatic analyses used in the study are well outside my

expertise and I have not tried to critique these aspects of the manuscript. However, I do have expertise in the biology of reactive oxygen species and redox signaling and my review focusses on how the manuscript fits with my field.

The authors have used single cell mass cytometry to document changes in individual cell types under different conditions. This is valuable information, but where I have major concerns is the context in which it is being applied, ie in describing differences in terms of a “reactive oxygen species regulome”, which I consider to be based on assumptions that do not necessarily hold. One of my biggest concerns and disappointments is that the authors have chosen to relate their cytometry data to fluorescence measurements using probes that are well established in the redox field to be ill advised as they are open to artifact and to multiple interpretations. This is documented in the reference the authors quote (ref 22) and more explicitly in the Guidelines for Use Consensus Statement by Murphy et al in Nature Metabolism (DOI: 10.1038/s42255-022-00591-z). These shortcomings are particularly relevant for the way in which the probes were used in this study.

Main concerns

1. Experiments using the probes, DCF, mitoSOX and BODIPY with cultured cells (Supplementary Fig 1b and description from line 182). The methodology described for these experiments involved adding the dyes to cells and after 30 min and a wash, carrying out a single fluorescence reading. The inference is that they were measuring accumulated “ROS” resulting from an ongoing exposure to H₂O₂ but this is not the case. The authors describe their system in terms of duration of H₂O₂ treatment of their cells (0.5, 4 and 48 h), but the H₂O₂ would have been consumed before their first probe analysis. So they were looking at cells that may have adjusted to a new state as a result of the stress.
2. At best these probe assays can give a measure of some form of redox activity during the period they are incubated with the cells. But for such assessment, it is essential to measure of a change in fluorescence over the period of analysis. A single point measurement, as in this study, will give a fluorescence signal that will depend on how much dye is taken up, how much comes out in the wash, as well as numerous factors, only some of which are redox-related. These determinants of fluorescence will also depend on cell type so I don't see how you can interpret data from 4 cell types together as appears to be the case here. It is also not possible to decipher how the different time points relate to the data presented or how the data were manipulated to give the -2 to +2 scale for the probes in Fig S1B. In all, there are serious limitations with the probe experiments. I cannot see that they have value as a measure of “ROS” levels and to try to do so detracts substantially from the manuscript.
3. It is well documented that treatment of cells with H₂O₂ does initiate a stress response that results in altered expression of a number of proteins including some involved in oxidant generation and removal. This, I assume, is what was measured in the current study. Such responses have been described in published RNAseq and proteomic studies. Results from these studies could be considered in more detail here and the cytometry measurements would be much better validated by relating to such changes. Fig 1D shows this relationship for absolute levels for 2 proteins, but what the manuscript needs is some correlation between changes in expression seen with single cell cytometry and another methodology. Expression changes in response to H₂O₂ are not necessarily the same for different cell types and the data acquired here could be usefully applied to probe such differences.
4. Although a set of protein expression changes can be identified as a response to H₂O₂, the reverse, that if such changes are observed they are the result of exposure to H₂O₂ (or ROS) does not necessarily follow. The latter, erroneous, assumption is the basis of the main theme of the current study and the use of the term redox regulome in this context. A number of stresses or stimulants give cellular responses and changes in protein expression that overlap with H₂O₂. Some are oxidative, some are not. Therefore. It is likely that a change in expression of one or more of the proteins followed in this study may or may not be due to an oxidative event. Also, it is likely that there are a host of reasons why levels of expression of different proteins, including oxidant-responsive proteins, vary in different cells. These levels are likely to be relevant as a basis when considering redox responses of the cells, but it would be overly simplistic to attribute such differences to oxidant exposure and the activity of a a redox regulome.
5. There are a lot of places where results are described with minimal explanation of how the data were obtained and manipulated before presentation. Units or parameters are often not defined.
6. I may have missed it but I could not find a list of antibodies and their sources.
7. Results using an antibody to oxidized PTPs are described (see line 284). A source and validation of this should be given. Is it the only one that I am aware of, an ScFv conformation sensing antibody produced by the Tonks group. I would like to see it validated in this setup.
8. O2.- should have a subscript (Introduction).

Overall, I consider that this is an exciting technique with considerable potential. The vast body of information provided in the manuscript is valuable in documenting expression levels of antioxidant proteins and proteins loosely associated with oxidative responses in differently treated immune cells. However, I consider there are major shortcoming in how the changes have been linked to oxidative stress, and I am not convinced with the fundamental concept of the manuscript that they represent a direct readout of reactive oxygen species that represents a “ROS regulome”.

Reviewer #4

(Remarks to the Author)

The manuscript by Wang et al. presents an innovative approach, termed single-cell reactive oxygen species (ROS) regulome profiling (scROP) to map the redox dynamics in immune cells at the single-cell level. The study is well-conceived and addresses a significant gap in understanding how redox states influence immune cell function, with relevant examples in the context of T cell activation, CAR-T cell persistence, and the effects of hypoxic conditions. However, there are a few areas where clarification, additional data, or methodological adjustments are necessary to strengthen the conclusions drawn.

Overall, I believe that the manuscript is a strong contribution to the field, offering a novel technique with significant potential. Addressing the comments below will help to clarify the study's impact and ensure that the findings are both reliable and broadly applicable.

Major points:

ROS modeling with H₂O₂ and ROS indicators (mitosox, H₂DCFDA, BODIPY):

The manuscript starts with a large-scale effort in identifying the most suitable antibodies for determining single-cell ROS states. While MitoSOX, H₂DCFDA, and BODIPY are standard indicators, their correlation with H₂O₂ concentration and duration needs to be better substantiated with data. A comparison of how these indicators correlate with each other under identical conditions would strengthen the methodology section, ensuring that these tools accurately reflect the ROS conditions in cells.

Selection of ROS markers and clones:

Related to the above point, I think this manuscript could be a comprehensive resource for others if it were to include more antibody QC data. The selection of the final 25 ROS markers seems somewhat arbitrary without sufficient explanation. Including a more comprehensive rationale for selecting these markers, along with data for all tested clones, would provide transparency and allow readers to understand the decision-making process.

scROP vs ROS indicators:

Including more data on the relationship between traditional ROS indicators and the scROP antibodies might also help to better demonstrate how scROP goes beyond the use of the existing indicators (which, one could argue, can already be used to determine single-cell ROS states).

T cell exhaustion versus activation:

The study's claim of T cell exhaustion in Figure 5 is questionable given the short time frame and in vitro conditions. T cell exhaustion typically requires chronic stimulation, which is difficult to replicate in vitro. The authors should consider reinterpreting these findings as prolonged activation rather than exhaustion unless additional functional data, potentially from in vivo models, can support the exhaustion claim. This would add credibility and depth to their conclusions.

Detailed discussion of limitations:

The authors should include a more thorough discussion of the limitations of their study, including the challenges of interpreting redox data, the potential for variability in antibody-based detection, and the need for further validation in different biological contexts.

Minor points:

In Figure 1g, the manuscript should comment on the group of cells that appear to be of mixed lineages (left of basophiles). What are these cells and why were they grouped together?

For Figure 2cde, the manuscript should clarify the number of donors shown (I guess it is one?) to ensure the robustness of the findings.

The rationale for selecting 15 metaclusters in Figure 3 should be clearly explained, including the criteria used for defining these clusters. In addition, there appears to be some inconsistency in terms of what is called a cluster vs. a metacluster.

There is only a minor difference in pimonidazole staining between normoxic and hypoxic conditions in Supplementary Figure 6, which is surprising and should be discussed.

The manuscript should offer a clearer explanation of what the CytoScore represents, including how it is calculated. The manuscript talks about localization, which I don't think is quantified here.

Version 1:

Reviewer comments:

Reviewer #2

(Remarks to the Author)

The Authors have sufficiently addressed my concerns.

(Remarks on code availability)

Reviewer #3

(Remarks to the Author)

As noted in my original assessment, I consider this a sophisticated and powerful methodology, that with appropriate qualifications and cautions is a useful addition to the literature. It provides information on expression changes in cells treated with H₂O₂, and as a method for comparing treatments has the potential for further applications. However, I remain unconvinced that the pattern of changes can be taken as a specific indicator of an oxidative stress response. I reiterate that although a pattern of change may occur in response to an oxidant, the reverse conclusion that such a change is specifically due to an oxidant does not necessarily hold. In this respect the additional experiment with IL4 is helpful but does not overcome the problem. Therefore, for me to be satisfied, I would need a big change of emphasis in the context and conclusions. It would be a pity not to publish this extensive study, but in my opinion it needs to be heavily revised and presented in a much more moderate fashion, as a generalized signaling assay with some relationship to oxidative stress. I would not recommend it in its current context as a readout of a specific reactive oxygen species regulome.

I also still have major concerns about specific points from my original review.

Points 1 and 2. Use of fluorescent "ROS" probes. Despite the authors' justification, information obtained from using these probes in the way they have done is essentially uninterpretable and meaningless. As I explained previously, I cannot see how making a single reading with an oxidant-sensitive probe added many hours or days after adding peroxide (which would be consumed in minutes) can be taken as a measure of ongoing oxidative stress. Understanding changes in response between adding the probe at 4 and 48 hours (new fig s4), and interpreting differences between cells in terms of oxidative stress are equally problematic. For a rigorous journal such as Nature Communications, these assays as they stand should not be used as a validation of their method and need to be removed from the submission.

Point 7. The authors have based their PTP oxidation findings on use of an antibody that apparently recognizes oxidation to the sulfonic acid. The apparent detection of PTPs oxidized to the sulfonic, (or sulfinic) acid under the conditions of this study is surprising as it would be expected to take a higher peroxide concentration exposure to the cells than used in this study (see for example Io Conte & Carrell JBC 2013). If any proteins formed sulfinic/sulfonic acids it would be peroxiredoxins, making the antiPrdxSO₂ antibody a more effective detector. Was this the case?

(Remarks on code availability)

I note the authors describe the availability of the codes they used but I am not an expert in this area so have not looked into it.

Reviewer #4

(Remarks to the Author)

In their revised version of the manuscript, the authors have satisfied most of my initial comments. Importantly, they have included their antibody QC data, correlations with traditional ROS markers and new data to support their claim of T cell exhaustion. Some minor questions remain around the topics of antibody selection, correlation of ROS markers with each other (to show their robustness), and the reproducibility of some new data. Overall, I do think that the manuscript now provides an improved demonstration of the capabilities and limitations of the presented method.

Remaining comments:

The inclusion of the H₂O₂ treatment effects on ROS indicators and the heatmap of correlation values of these indicators with the tested antibodies helps to make their antibody selection process more transparent. However, I still do not quite understand how the 25 final markers were selected. Was this just the X top ranked antibodies in each correlation modules?

Related, do I understand correctly that H₂DCFDA, BODIPY, and MitoSOX were measured simultaneously on the same cells? That would be important to also correlate these indicators with each other on a single cell level to assess how robustly these markers, which are here used as a gold-standard, reflect ROS conditions in the cells.

The inclusion of the LCMV exhaustion experiment is a worthy addition to the manuscript. I do want to point out though that the observed differences in the scRIP markers are minute and on the edge of what I would consider detectable by CyTOF. It would be important to show that this difference can be robustly observed when looking at biological replicates.

Related to Figure 2cde, I think that showing the pseudotime analysis for the other 2 donors (in the supplements) as was performed for the representative donor in the supplements would ensure consistency.

(Remarks on code availability)

Version 2:

Reviewer comments:

Reviewer #3

(Remarks to the Author)

The authors have taken on board my concerns and I am satisfied that they have made appropriate changes. I find the manuscript acceptable in its revised form.

(Remarks on code availability)

Reviewer #4

(Remarks to the Author)

The authors have now addressed my previous comments and questions, thank you.

(Remarks on code availability)

Reviewer #1 (CyTOF) (Remarks to the Author):

In this manuscript, the authors investigate reactive oxygen species at the single cell level that they call a single-cell ROS regulome profiling method. They initially carry out experiments in individual cell lines to develop their targets. Then in blood samples from healthy donors, finally in patients with medical conditions: kidney dialysis, hemophilia, blood cancers. I found the paper fascinating and readable, but my knowledge of biology is not at a level where I can make meaningful comments on these studies.

We sincerely thank this reviewer for the thoughtful and constructive review of our manuscript. We were pleased to hear that the reviewer found the study both fascinating and readable. We have responded to each comment as detailed below and believe that our changes and additions to the manuscript and supplemental material address the concerns raised.

Methods. In the Methods section there is no information provided about antibody labeling for CyTOF measurements. In the Introductions and Results section, the authors write “We first conducted a thorough screening of more than 100 antibodies targeting redox-related proteins.” “From this vast screening, 25 antibodies were chosen for their strong correlations with standard ROS measures.” Presumably these are the antibodies stained with Maxpar reagents (Fluidigm/Standard BioTools) for their CyTOF experiments. Even more curious, the term Maxpar does not appear anywhere in the manuscript.

We apologize that the antibody labeling process was not clearly described. We have now revised the Methods section to include detailed information on the metal conjugation of antibodies using the Maxpar X8 Antibody Labeling Kit (Fluidigm/Standard BioTools). The updated text follows (lines 1137-1144):

“Metal conjugation of antibodies for CyTOF analysis.

Primary conjugates of mass cytometry antibodies were generated using the Maxpar X8 Antibody Labeling Kit (Fluidigm/Standard BioTools) following the manufacturer’s instructions. After labeling, the antibodies were diluted to a concentration of 0.2–0.5 mg/mL in Candor PBS Antibody Stabilization Solution (Candor Bioscience GmbH) containing 0.02% NaN₃ and stored at 4°C for long-term use. Each antibody clone and lot was titrated to determine optimal staining concentrations using appropriate positive and negative controls.”

Tables at the end of the manuscript identify which metals are on each antibody. Several of the Tables have no title. There is a Supplementary Table 3 Supplementary Table 2 or 1.

We apologize for the oversight regarding the table titles. In the revised version, all tables are properly labeled with appropriate titles.

Reviewer #2 (ROS, redox) (Remarks to the Author):

This manuscript by Wang et al. develops a new single cell assay (scROP) that attempts to bring redox biology to the single cell arena and applies the assay to CAR-T therapy in patients. The potential is highly significant, since ROS and redox-focused assays are technically challenging they are rarely comprehensive nor amenable to single cell analysis. The approach taken by the authors is ambitious, and the assay itself is a significant advance in the field. Furthermore, their application to CAR-T patients is also ambitious, and has high clinical relevance, and clearly interesting patterns emerge from the scROP results demonstrating its utility. However, despite these strengths, there are several major concerns detailed below.

One major concern is related to accuracy of the claims, as this assay is not comprehensive, nor is there is their clear evidence that the assay is even specific for redox biology. scROP is more of a generalized signaling assay that incorporates some minor input from ROS and redox biology, but most of these markers are also regulated by non-redox stimuli and the ROS-specific markers do not appear to be terribly informative in the data presented. Thus, the scROP assay is useful, but the relevance of redox biology per se isn't clear and the specificity for redox processes is quite unlikely. A second major concern is that nearly all the claims are associative or correlative, with very limited functional studies aside from inhibition of REF-1 that was driven by knowledge of redox biology as opposed to being revealed by the scROP assay results themselves. Even in the abstract, the phrases used are vague, including "established a connection between", "linked", and "significant correlation".

We appreciate this reviewer's insightful comments and constructive feedback. We are encouraged by recognition that our scROP assay will advance the field of redox biology. Your concerns, including those regarding the specificity and comprehensiveness of the assay, have been addressed them in this revision.

Main concerns:

1. The authors routinely claim they are 'comprehensively' measuring the 'ROS regulome', for example line 185 states "To investigate redox states comprehensively, we aimed to quantify the abundance of ROS transporters, pivotal ROS-generating and ROS-scavenging enzymes and their regulatory modifications (e.g., phosphorylation), products of prolonged oxidative stress (e.g., sulfonic oxidation modification of proteins), and the transcription factors and signaling molecules that drive specific redox programs, collectively termed the cellular ROS regulome" Yet, they aren't measuring any NOX, only 1 of 6 PRDXs, and only a single Aquaporin. Thus the data provided isn't comprehensive, and wouldn't be sufficiently complete to understand the architecture of ROS production. Moreover, they claim on line 167 that they've developed a "panel that allows comprehensive

exploration of how oxidative stress is generated and neutralized as well as impacted mechanisms” which is vastly overstated based on the limited markers assayed. This assay is nowhere near capable of ‘omics’, and claiming that this assay can measure anything close to a regulome need to be dramatically reduced.

We acknowledge that some of our language may have overstated the assay’s scope, and we have revised our manuscript to clarify its focus and limitations. For example, we now state (lines 229-231): *“The markers selected for the scROP panel enabled us to trace key redox dynamics involved in the activation of T cells, identifying significant alterations within the ROS network.”* Additionally, we revised text in the Results section to emphasize the targeted nature of our analysis (lines 244-245): *“To investigate the interconnected signaling pathways involved in redox regulation, we aimed to quantify the abundances of ROS transporters, pivotal ROS-generating and ROS-scavenging enzymes and their regulatory modifications (e.g., phosphorylation), products of prolonged oxidative stress (e.g., sulfonic oxidation modification of proteins), and the transcription factors and signaling molecules that drive specific redox programs, collectively termed the cellular ROS regulome.”*

With regards to antibody selection, we initially screened over 100 antibodies, covering a broad range of ROS-related molecules including NOX1, NOX2, and PRDX1-PRDX5 (see Supplementary Table 1). However, some of these markers did not meet the selection criteria for H₂O₂-based experiments due to inconsistent or nonspecific results. The absence of specific NOX or PRDX proteins may reflect regulatory factors or context-dependent activities unique to different immune cell types, as observed in previous studies^{1,2}. These variations suggest that the effects of H₂O₂ on ROS-related proteins are highly dependent on the cellular environment and conditions.

2. Relatedly, nearly all of the antibodies chosen are not redox specific. Phospho-ERK and phospho-AKT, and most protein expression, are also affected by many non-redox perturbations. It is therefore unclear much influence redox biology has on the observed changes as opposed to non-redox-dependent change. For example, how much of the time-dependent changes observed by scROP in Figure SA really depend on redox biology? It seems to be low since no redox-specific marker (e.g. oxDJ1, oxPTP) is highlight in the next panel, Figure 2B. Insight into how much redox biology influences this could be gleaned comparing scROP assay results upon addition of an antioxidant, but otherwise the results d not demonstrate that the scROP assay is highly specific for redox signaling / biology. This is a major concern since the assay is positioned to measure redox biology, even the redox ‘regulome’, and re-positioning away from a narrow focus on redox biology is needed. For example, many of the observed changes in Fig. 2C happen before any increase in oxidization of DJ1 or oxPTP, which would presumably measure the onset of ROS production, clearly indicating non-redox-dependent change are detected by this assay.

Similarly, the 'active' versus 'basal' types of CAR-positive T cells (Fig. 3B), are really driven by 4 markers, protein expression of NNT, GPX4, HSP70 and phosphorylation of p38MAPK. This is hardly a change in the redox regulome, and it's hard to claim that these markers are even redox-specific. In short, the scROP assay is useful, and may include some aspects of redox biology, but its specificity to redox biology is very inconclusive.

We understand your concerns about the influence of non-redox factors on our findings, and we have conducted additional analyses to address these concerns. First, to address the concern that many of the observed protein expression changes could be driven by non-redox factors, we conducted control experiments in which SM826 cells were treated with IL4, a well-established non-oxidative stressor ³. We observed no significant differences in MitoSOX and MitoScore metrics between the IL4-treated and untreated groups. This result suggests that the protein expression changes observed in response to H₂O₂ are indeed specific to oxidative stress and are not driven by general cellular stress responses. This finding is discussed in the text (lines 319-321): *"In contrast, treatment of cells with IL4, a non-oxidative stressor, did not result in a change in the MitoScore, demonstrating that the observed effects were specific to oxidative stress."*

Second, we evaluated specificity of redox markers in 'active' versus 'basal' CAR-positive T cells. We appreciate the reviewer's concern regarding the heatmap in old Fig. 3b, which represents averaged data and may obscure marker-specific variations due to potential outliers. In response to this concern, we have reanalyzed the data and plotted the expression levels of individual markers in new Supplementary Fig. 10 and have categorized them into 15 clusters based on their redox profiles using FlowSOM analysis as shown in new Supplementary Fig. 9. This reanalysis revealed that three specific redox-related markers, oxPTP, PDI, and Ref/APE1, consistently separated the active and basal CAR-positive T cells. These markers are significantly differentially expressed between the two states, confirming their roles as redox-specific indicators.

To further elucidate the differences between the active and basal states, we generated bar plots comparing the expression of these redox markers and other features across the basal and active metaclusters (new Fig. 3c). The bar plots provide clear evidence of the functional differences in redox biology between the two states. We believe that these new analyses offer a stronger and clearer perspective on the specificity of the scROP assay in capturing redox biology.

Finally, we applied the scROP panel to examine redox profiles in cells treated with APX2009, N-AC, or their combination. The scROP analysis revealed that APX2009 treatment disrupted key redox correlation networks, including Ref/APE1-ERO1B and GPX4-ERO1B-pNFkB (new Supplementary Fig. 18). Notably, combination treatment with N-AC restored these disrupted networks, providing direct evidence of the redox dependency underlying the observed effects. These findings underscore the capability of the scROP approach to capture redox-specific changes at the single-cell level.

3. The authors state that they ‘validated’ their assay in Figure 1, claiming that their MitoScore (an expression measure) matches levels of bona fide mitochondrial ROS levels using mitoSOX. However, the validation criteria are not stated, and a simple experiment generating mitochondria ROS and observing an increase in their mitoROS proxy (MitoScore) would be excellent validation. However, this isn’t shown and it’s unlikely that mitochondrial protein expression correlates with mitochondrial ROS. Additional validation is needed to make claims about being able to assess mitochondrial Ros using this assay.

This comment prompted us to further clarify the distinction between real-time mitochondrial ROS detection using MitoSOX and the sustained mitochondrial response measured by MitoScore. In these new experiments, we treated the SM826 cell line with pyocyanin ⁴, a redox-active metabolite produced by *Pseudomonas aeruginosa* that disrupts mitochondrial respiration and impairs energy production. After 48 hours of stimulation, we observed a significant increase in MitoScore (new Fig. 1f; see Supplementary Table 2), indicating a sustained mitochondrial redox response. In contrast, control experiments using IL4 ³, a non-oxidative stressor, showed no such changes, validating that the observed effects were specific to oxidative stress. We attribute the difference between MitoSOX and MitoScore to distinct mechanisms of detection: MitoSOX provides real-time detection of mitochondrial superoxide, reflecting the immediate response to oxidative stress. In contrast, MitoScore captures overall expression related to mitochondrial function, which takes longer to show significant changes, thus highlighting a more sustained mitochondrial redox response.

The text has been revised to discuss these findings (line 315-321): *“To confirm that the responses captured by scROP are specific to redox activity, we stimulated the SM826 cell line with pyocyanin, a compound known to disrupt the redox functions of complex III and to induce mitochondrial superoxide production. As expected, after 48 hours of stimulation, the MitoScore increased significantly (Fig. 1f and Supplementary Table 2), indicating a sustained mitochondrial redox response. In contrast, treatment of cells with IL4, a non-oxidative stressor, did not result in a change in the MitoScore, demonstrating that the observed effects were specific to oxidative stress.”*

4. The functional utility of the scROP assay isn’t established. While the authors use a REF-1 inhibitor in figure 5, this choice isn’t driven by scROP assay results. Nearly all the claims are associative or correlative, with very limited functional studies aside from inhibition of REF-1 that was driven more by knowledge of redox biology as opposed to being driven from scROP assay results. Even in the abstract, the phrases used of “established a connection between”, “linked”, and “significant correlation”. The descriptive nature of these studies limits their significance.

Thank you for this insightful feedback. We understand your concern that the experiment with the REF-1 inhibitor APX2009 was driven by prior knowledge rather than scROP assay results. In our experiments, we utilized APX2009 to explore the causal link between early redox equilibrium and T cell effector functions. Although it is true that prior

knowledge of redox biology informed the choice of APX2009, the scROP assay played a crucial role in guiding our experimental approach and in providing key mechanistic insights into the redox changes that occurred upon APX2009 treatment. Specifically, we applied the scROP panel to compare redox profiles of cells treated with APX2009, N-AC, or the combination. The scROP analysis revealed that APX2009 treatment disrupted critical redox correlation networks, such as the connections between Ref/APE1-ERO1B and GPX4-ERO1B-pNFkB (new Supplementary Fig. 18). Combination treatment with N-AC restored these networks, which directly demonstrated the redox-dependency of the observed effects. The scROP assay provided mechanistic insights into how Ref/APE1-mediated redox equilibrium influences T cell exhaustion, which would not have been possible without the scROP assay's ability to capture detailed changes in the redox regulome at the single-cell level. Furthermore, the functional outcomes observed in the APX2009-treated cells, such as a significant decrease in TNF α - and IFN γ -producing CD8⁺ T cells, were validated using the scROP assay, where disrupted redox regulation was shown to impair mitochondrial function and drive T cell exhaustion (Fig. 5e, f). These findings underscore the functional utility of the scROP assay in revealing the role of redox regulation in T cell functionality.

We hope that this explanation provides greater clarity on the value and utility of the scROP assay in driving our experimental approach and functional conclusions. To address this in the text, we now state (lines 563-573): *“To gain deeper mechanistic insights into how Ref/APE1-mediated early redox equilibrium impacts T cell exhaustion, we applied the scROP panel to compare APX2009-treated cells, N-AC-treated cells, and cells treated with the combination. APX2009 treatment disrupted key redox correlation networks as we observed the loss of connections between Ref/APE1-ERO1B and GPX4-ERO1B-pNFkB (Supplementary Fig. 18). Interestingly, these disrupted links were restored by the addition of N-AC, suggesting that APX2009 interferes with the coordination of the ROS regulome, particularly interactions between the endoplasmic reticulum and mitochondrial components such as GPX4. Consistent with this, mitochondrial fitness, measured using MitoTracker Deep Red, was reduced following APX2009 treatment, indicating that disrupting Ref/APE1-mediated redox equilibrium can impair mitochondrial function and drive T cell exhaustion (Fig. 5f).”*

We also appreciate your feedback regarding the phrasing in our abstract. We understand your concerns regarding terms such as "established a connection" and "significant correlation" and have revised the abstract to better reflect the concrete, experimentally validated outcomes and the implications of our findings in a more direct manner. Below is the updated version (lines 152-160): *“scROP quantifies ROS transporters, enzymes, oxidative stress products, and associated signaling pathways to provide information on cellular redox regulation. Applied to diverse cell types and conditions, scROP revealed unique redox patterns and dynamics including coordinated shifts in CD8⁺ T cells upon antigen stimulation and variations in CAR-T cell persistence. Furthermore, scROP highlighted the impact of environmental factors such as hypoxia on redox balance and T cell exhaustion and identified distinct redox features in hemodialysis patients. These findings underscore the power of scROP to elucidate intricate redox networks and their implications in immune cell function and disease.”*

Minor points

- The authors claim that Fig. 1F shows that each type of normal immune population has a “discovered that each cell type has a unique redox pattern”, but B and T cells look quite similar, as do basophils and NK cells. More rigorous analysis is needed to make this claims, especially since Fig 1g shows that the scROP pattern of many of these immune cells overlap.

In response to the concern regarding the similarity of redox patterns among different immune cell populations, we have implemented a more rigorous gating strategy to enhance the specificity of our analysis. We now examine 18 distinct immune subtypes, for example, separating T cells into CD4⁺ and CD8⁺ subsets and further distinguishing them into naïve and effector cells as well as differentiating NK cells into CD56^{bright} and CD56^{dim} subsets. This refined approach revealed unique redox characteristics among the various immune populations. Notably, B cells exhibited similar ROS expression levels to CD4⁺ naïve and effector cells. In contrast, CD8⁺ naïve and effector T cells and central and effector memory T cells have significantly elevated expression levels of SOD2(K68Ac), ERO1B, and PDI compared to other cell types. Additionally, CD56^{bright} and CD56^{dim} NK cells express different amounts of SOD2(K68Ac) and ERO1B, indicating unique redox regulation within these NK populations. These findings highlight the necessity of analyzing immune cell subsets to uncover distinct redox signatures that are critical to their functional roles. These new results are shown in Fig. 1g of the revised manuscript and are reproduced below.

Fig. 1g. Heatmap of ASINH transformed mean expression levels of all evaluated ROS regulome factors across various immune cell lineages.

- Details of the machine learning model are insufficient. The authors state that “The most accurate algorithm, Catboost, was selected as the main algorithm for training the model.”, but how was this accuracy assessed and was this based on training data, testing data, etc? Also, on what basis did the authors decide that “interindividual variability of ROS markers was assumed to be negligible”? Was

the testing performed for only a single random state and random assignment of samples to Test? If so, that is insufficient. Lastly, it is unclear how the authors went from a correlation (“Spearman correlation coefficient between the predicted dialysis time (in months) and the actual dialysis time”) to the accuracy and ROS plot shown in Fig. 1H.

As we described in the Methods section on machine learning methods, the accuracy of the CatBoost algorithm was rigorously assessed through 10-fold cross-validation, which involved partitioning the dataset into 10 folds. In each iteration, one fold was used as the testing set, and the remaining nine folds served as the training set. This process was repeated 10 times, with each fold being used once as the testing set. The model's performance was evaluated using metrics such as the Spearman correlation coefficient between the predicted and actual dialysis times. This method not only mitigates the risk of overfitting but also ensures robustness and generalizability by validating the model across multiple random states and data splits. Furthermore, CatBoost was selected as the primary model based on its highest macro-average F1 score, as detailed in new Supplementary Table 15. This table summarizes F1 scores for categories 0–5 and includes accuracy, macro-average, and weighted-average metrics. Models were ranked by macro-average F1 scores, confirming CatBoost's superior performance and balanced generalizability across all classifications during cross-validation.

The assumption of negligible interindividual variability in ROS markers was based on our preliminary analysis of the study population: There was a relatively homogeneous distribution of ROS marker levels across participants. This homogeneity, along with our focus on developing an initial predictive model, led us to simplify the model by not accounting for interindividual variability at this stage. However, we acknowledge that interindividual variability can significantly impact model performance.

To enhance the reliability of our model, we employed multiple random assignments of samples to both the training and testing sets, which contributed to a comprehensive evaluation of the model's performance. The Spearman correlation coefficient was utilized to quantify the relationship between predicted and actual dialysis times, reflecting the model's predictive capability. The accuracy metrics and the Receiver Operating Characteristic plot presented in Fig. 1i were derived from this correlation analysis, illustrating how well the model can predict dialysis duration based on ROS features. We have clarified this transition in the manuscript to highlight the relationship between correlation analysis and the model's predictive accuracy. We now state (lines 1314-1317): *“Specifically, we employed 10-fold cross-validation with 10 runs, using different random states for each run. This allowed us to evaluate the robustness and consistency of the model across multiple random states.”*

Additionally, Fig. 2i demonstrates the predictions generated by an *in vitro*-trained model applied to *in vivo* data. Specifically, when we analyzed the *in vivo* day 7 (early) and day 14 (late) groups, kernel density estimation (KDE) fitting was used to visualize the distribution of predicted days. The KDE fitting revealed distinct patterns: the *in vivo* day 7 group predominantly clustered near the *in vitro* day 0 predictions (orange), while the *in*

in vivo day 14 group shifted toward the *in vitro* day 5 predictions (blue). This indicates that the model effectively captures temporal ROS dynamics, with earlier *in vivo* time points aligning with earlier *in vitro* predictions and later time points showing a corresponding shift. These results underscore the model's ability to distinguish ROS dynamics at different time points, highlighting its potential utility in predicting dialysis duration based on ROS markers. To further substantiate the robustness and reproducibility of our predictive model, we have included the triplicate training data used for model evaluation, provided as Fig. R1, which demonstrate consistent performance and reinforce the validity of the *in vitro*-trained model in accurately interpreting ROS-related temporal changes.

Fig. R1. Predictive performance of the *in vitro*-trained model on *in vivo* data

The figure illustrates the predictive outcomes of an *in vitro*-trained model applied to *in vivo* data for day 7 (early) and day 14 (late) groups.

Top panel: Kernel density estimation (KDE) plot showing the distribution of predicted dialysis days. The orange curve represents the *in vivo* day 7 group, with predictions clustering predominantly near *in vitro* day 0, while the blue curve represents the *in vivo* day 14 group, with predictions shifting toward *in vitro* day 5. This demonstrates the model's ability to distinguish temporal differences in ROS dynamics between the two groups. Bottom panel: Stacked bar chart showing the probability distribution of predicted dialysis days for *in vivo* day 7 (orange) and day 14 (blue) groups. The frequencies indicate a clear temporal shift in predictions, further validating the model's capability to differentiate between groups based on ROS features.

• **Given the importance of the MitoScore to validation of this assay, it would be useful to see how measure that changes over pseudotime in the relevant figures and machine learning-based classifications.**

In response to the request for insight into the changes in MitoScore over pseudotime, we conducted a new comparative analysis of both MitoScore (from CyTOF) and MitoSOX

(from flow cytometry) to assess their dynamic behaviors throughout pseudotime. Our analysis demonstrated that both measures exhibit robust and dynamic behavior. Notably, we observed that MitoSOX increased significantly from day 2 and remained high thereafter. Additionally, MitoScore increased around pseudotime 0.4. MitoScore and MitoSOX provide complementary insights into mitochondrial ROS levels and enhance our understanding of mitochondrial function, and these data supporting the reliability of our assay in capturing changes in redox states throughout the pseudotime continuum.

These data are shown in new Fig. 2e (reproduced below), and the following has been added to the text (lines 383-387): “Over pseudotime, both MitoScore and MitoSOX metrics were dynamic (Fig. 2e). MitoSOX significantly increased from day 2 and MitoScore peaked around pseudotime 0.4. These findings provide complementary insights into mitochondrial ROS levels, reinforcing the reliability of our assay in capturing redox state changes throughout the pseudotime continuum.”

Fig. 2e. Violin plot of normalized MitoSOX values from day 0 to day 5 ($n=3$ independent samples). Red line is MitoScore presented as the 99th percentile normalized values over pseudotime, smoothed using a window size of 1000 ($n=2$ independent samples, data from one representative sample shown).

• **The result (Fig. 2G) that protein folding and translation happens well before activation of most pathways and “Cellular response to ROS” along with the notable delay in MAP kinase activity is very interesting and a unique insight that this assay could provided. This should be discussed, especially since it seems like MAPK-dependent activation by direct oxidation (the canonical model) doesn’t appear to be occurring.**

Our data do indicate that protein folding and translation occur well before the activation of most pathways, with a notable delay in both the "Cellular response to ROS" and "MAP kinase activity" (old Fig. 2g, now Fig. 2h). We have therefore expanded the discussion to highlight how these findings challenge the canonical model of MAPK activation via direct oxidation. We have expanded the discussion as follows (lines 649-659): “The traditional view posits that ROS directly oxidize MAPKs (e.g., ERK⁵ and p38⁶) leading to their rapid activation. However, the delay in MAPK activation observed in our study suggests that this process may be regulated through indirect mechanisms, such as secondary signaling pathways, post-translational modifications, or redox-sensitive intermediaries, rather than

through immediate oxidative modifications. This temporal dissociation indicates that metabolic reprogramming and mitochondrial redox signals may play more critical roles in modulating MAPK activity than previously thought. Our results emphasize the complexity of redox signaling in immune cells and suggest that the classical model of MAPK activation should be reassessed. Further work will be needed to elucidate the mechanism that drives the interplay between mitochondrial redox dynamics and T cell signaling pathways.

- **The authors state that Fig. 2F “identified four distinct groups of features with analogous correlation patterns”, but this isn’t clearly indicated.**

To clarify, we have revised the legend for old Fig. 2f, now Fig. 2g, to provide a more detailed explanation of how the four distinct groups are identified and the correlation patterns observed (lines 831–838). *“ASINH transformed data of OT-I CD8⁺ T cells annotated with GO biological processes. Red represents a positive correlation, and blue represents a negative correlation (n=3 independent samples, data from one representative sample shown). The features are grouped into four categories based on their functional roles: kinase signaling (EOMES, TIM3, oxPTP, pERK); protein synthesis/translation (HSP70, TCF1/7, REF/APE1, and GR); DNA damage/peroxidation (NNT, 53bp1, AQP8, PD1, CD137, Catalase, and ACOX3); and anti-oxidation (CD62L, oxDJ1, pNFkB, ERO1B, CTLA4, QSOX1, LAG3, KEAP1, and GPX4).”*

Reviewer #3 (ROS chemistry) (Remarks to the Author):

This manuscript describes an enormous undertaking in which the authors have used the newish and powerful technique of mass cytometry to examine proteomic changes in single cells. While there are other methods that can provide similar information for a population of a particular cell type, a major advantage of this technique is that it can be applied to heterogeneous cell populations. The technique and the bioinformatic analyses used in the study are well outside my expertise and I have not tried to critique these aspects of the manuscript. However, I do have expertise in the biology of reactive oxygen species and redox signaling and my review focusses on how the manuscript fits with my field.

The authors have used single cell mass cytometry to document changes in individual cell types under different conditions. This is valuable information, but where I have major concerns is the context in which it is being applied', ie in describing differences in terms of a "reactive oxygen species regulome", which I consider to be based on assumptions that do not necessarily hold. One of my biggest concerns and disappointments is that the authors have chosen to relate their cytometry data to fluorescence measurements using probes that are well established in the redox field to be ill advised as they are open to artifact and to multiple interpretations. This is documented in the reference the authors quote (ref 22) and more explicitly in the Guidelines for Use Consensus Statement by Murphy et al in Nature Metabolism (DOI: 10.1038/s42255-022-00591-z). These shortcomings are particularly relevant for the way in which the probes were used in this study.

We appreciate your recognition of the value of using mass cytometry to study changes in single cells, especially in mixed cell populations. We understand your concerns about using the term "reactive oxygen species regulome" and the limitations of the fluorescence probes in our study. We appreciate your reference to the consensus statement by Murphy et al. and have taken this into account in our revisions. We believe that our use of single-cell mass cytometry offers valuable insights into redox biology. We trust that we have addressed your concerns by providing more context about our use of fluorescence measurements and by improving our discussion on the implications of our findings.

Main concerns

1. Experiments using the probes, DCF, mitoSOX and BODIPY with cultured cells (Supplementary Fig 1b and description from line 182). The methodology described for these experiments involved adding the dyes to cells and after 30 min and a wash, carrying out a single fluorescence reading. The inference is that they were measuring accumulated "ROS" resulting from an ongoing exposure to H₂O₂ but this is not the case. The authors describe their system in terms of duration of H₂O₂ treatment of their cells (0.5, 4 and 48 h), but the H₂O₂ would have been consumed before their first probe analysis. So they were looking at cells that may have adjusted to a new state as a result of the stress.

We agree that the H₂O₂ would have been consumed or dissipated before the probe analysis. Consequently, the fluorescence readings do not represent the ongoing presence of H₂O₂ but rather reflect the cellular oxidative state and adaptations resulting from the prior oxidative stress. Our primary objective in this experiment was to assess the cellular response to oxidative stress induced by varying durations of H₂O₂ exposure. The methodology we used, involving H₂DCFDA, MitoSOX, and BODIPY probes, was designed to measure the cumulative oxidative changes and cellular redox state adjustments rather than real-time ROS production during H₂O₂ exposure. This approach allowed us to capture the downstream effects of oxidative stress on cellular components and their ability to adjust to a new equilibrium.

2. At best these probe assays can give a measure of some form of redox activity during the period they are incubated with the cells. But for such assessment, it is essential to measure of a change in fluorescence over the period of analysis. A single point measurement, as in this study, will give a fluorescence signal that will depend on how much dye is taken up, how much comes out in the wash, as well as numerous factors, only some of which are redox-related. These determinants of fluorescence will also depend on cell type so I don't see how you can interpret data from 4 cell types together as appears to be the case here. It is also not possible to decipher how the different time points relate to the data presented or how the data were manipulated to give the -2 to +2 scale for the probes in Fig S1B. In all, there are serious limitations with the probe experiments. I cannot see that they have value as a measure of "ROS" levels and to try to do so detracts substantially from the manuscript.

We acknowledge the limitations of single-point measurements with ROS probes, which can be influenced by factors such as dye uptake, efflux during washing, and non-redox-related determinants. These factors indeed vary depending on cell type and can confound the interpretation of results when used in isolation. The scROP assay employed in our study captures a broader and more comprehensive picture of redox activity than do traditional probe-based studies. The scROP assay simultaneously measures over 25 ROS-related proteins, offering a detailed profile of oxidative stress and its regulatory networks in individual cells. Our intention in including four distinct immune cell types (Raw264.7, SM826, SH-SY5Y, and HL-1, Supplementary Fig. 2-4) was to provide a comprehensive overview of redox dynamics in diverse cellular contexts. Each cell type represents unique metabolic profiles and responses to oxidative stress.

Traditional ROS probe assays provide only a snapshot of overall ROS levels, our scROP approach integrates these probes with a multi-parametric assessment, enabling a more reliable and detailed comparison of redox activity across multiple immune cell types. In our analysis, we evaluated distinct redox patterns that emerged during different treatments. Furthermore, our experiments were conducted over four time points (0, 0.5, 4, and 48 hours) and three different concentrations of H₂O₂ treatment (0, 10, and 100 μM), allowing us to capture the dynamics of ROS signaling in response to oxidative stress. This temporal and dose-dependent approach reveals how redox regulation evolves over

time, across different cell types, and under varying levels of oxidative stress. Compared to traditional ROS assays, this method provides deeper insights, as clearly demonstrated in new Supplementary Fig. 2 and 3 (scROP), in contrast to the limited findings shown in new Supplementary Fig. 4.

Regarding the -2 to +2 scale in Fig. S1b, this represents a Z-score normalization, which was applied to visualize the relative expression of ROS markers across different cell types. This normalization accounted for baseline differences in fluorescence intensity and marker expression and allowed for better cross-comparison between different immune populations. The time points used for analysis were consistent across all experiments, ensuring that the measurements reflect comparable stages of cellular activation and redox signaling.

We have revised the legend of Fig. S1b to better explain these methodological details and the robustness of the scROP assay, emphasizing its advantage over conventional probe-based ROS measurements as follows (lines 959-963): *"Pearson correlations between three representative antibodies linked to ROS and ROS indicators H₂DCFDA, BODIPY, and MitoSOX. Each point represents the mean average fluorescence, and Z-score normalization was applied to both antibody and ROS indicators expression values to standardize the data and facilitate comparison across different cell types and conditions."*

3. It is well documented that treatment of cells with H₂O₂ does initiate a stress response that results in altered expression of a number of proteins including some involved in oxidant generation and removal. This, I assume, is what was measured in the current study. Such responses have been described in published RNAseq and proteomic studies. Results from these studies could be considered in more detail here and the cytometry measurements would be much better validated by relating to such changes. Fig 1D shows this relationship for absolute levels for 2 proteins, but what the manuscript needs is some correlation between changes in expression seen with single cell cytometry and another methodology. Expression changes in response to H₂O₂ are not necessarily the same for different cell types and the data acquired here could be usefully applied to probe such differences.

In response to your feedback, we have now conducted a comparison between the fold-change expression levels obtained from published RNA sequencing data⁷ and the ASINH-transformed CyTOF data following H₂O₂ treatment across Jurkat cells. These data are shown in new Supplementary Fig. 6 (reproduced below) and are described in the text (line 309-312): *"Additionally, we compared the scROP profiling results with previously reported RNA-seq measurements⁷ in Jurkat cells, which further validated the relationship between RNA and protein expression levels in response to oxidative stress (Supplementary Fig. 6 and Supplementary Table 2)."*

Our analysis revealed a Pearson correlation coefficient of 0.55, indicating a moderate positive correlation between the two methodologies. This finding supports the validity of our cytometry measurements and demonstrates that the changes observed in protein expression levels measured by CyTOF are consistent with those seen at the transcriptomic level through RNA sequencing⁷. These findings underscore the utility of integrating single-cell cytometry with transcriptomic analysis in understanding the redox responses in various cellular contexts.

4. Although a set of protein expression changes can be identified as a response to H₂O₂, the reverse, that if such changes are observed they are the result of exposure to H₂O₂ (or ROS) does not necessarily follow. The latter, erroneous, assumption is the basis of the main theme of the current study and the use of the term redox regulome in this context. A number of stresses or stimulants give cellular responses and changes in protein expression that overlap with H₂O₂. Some are oxidative, some are not. Therefore. It is likely that a change in expression of one or more of the proteins followed in this study may or may not be due to an oxidative event. Also, it is likely that there are a host of reasons why levels of expression of different proteins, including oxidant-responsive proteins, vary in different cells. These levels are likely to be relevant as a basis when considering redox responses of the cells, but it would be overly simplistic to attribute such differences to oxidant exposure and the activity of a a redox regulome.

To address this concern, we conducted control experiments in which SM826 cells were treated with IL4, a known non-oxidative stressor³, specifically at the 48-hour time points. There were no significant differences in MitoSOX and MitoScore metrics between IL4-treated cells and cells not treated with IL4. This result suggests that the protein expression changes observed in response to H₂O₂ are likely specific to oxidative stress rather than general cellular stress responses. These data are shown in new Fig. 1f. These data further reinforce the idea that the changes we observed in response to H₂O₂ are primarily associated with redox regulation. We discuss this in detail in the revised manuscript (lines 319-321): *“In contrast, treatment of cells with IL4, a non-oxidative stressor, did not result*

in a change in the MitoScore, demonstrating that the observed effects were specific to oxidative stress.”

Although these results provide supporting evidence that the scROP panel is highly responsive to H₂O₂-induced oxidative stress, we acknowledge that it does not exhaustively cover all possible non-oxidative stress conditions. However, our results demonstrate that the scROP assay captures secondary and tertiary responses to oxidative stress, which, although not exclusively direct ROS responses, do represent cellular changes that are detectable following H₂O₂-induced oxidative challenge. We believe that this additional experiment strengthens our assertion that scROP is particularly suited to detecting redox-specific alterations in cellular states.

5. There are a lot of places where results are described with minimal explanation of how the data were obtained and manipulated before presentation. Units or parameters are often not defined.

In response to this comment, we have revised the manuscript to include the definitions of terms and parameters, along with the units of measurement. Additionally, we have clarified the data manipulation processes and specified sample sizes for each experiment. These updates aim to provide a clearer understanding of our methods and ensure the reproducibility of our findings.

6. I may have missed it but I could not find a list of antibodies and their sources.

The list of antibodies and their sources is provided in Supplementary Table 1 of the manuscript. This table includes detailed information on all antibodies used in the study, including their specific targets, sources, and catalog numbers. Please let us know if there are any further details or clarifications needed.

7. Results using an antibody to oxidized PTPs are described (see line 284). A source and validation of this should be given. Is it the only one that I am aware of, an ScFv conformation sensing antibody produced by the Tonks group. I would like to see it validated in this setup.

Thank you for highlighting the need for clarification regarding the oxidized PTP antibody used in our study. We used an antibody from R&D Systems (catalog #MAB2844, link), specifically selected for its ability to detect terminally oxidized cysteine residues, such as cysteine sulfonic acid (Cys-SO₃H). Unlike reversible oxidative modifications, terminal oxidation reflects irreversible changes that significantly alter protein structure, affecting charge and electron density within the protein's microenvironment. These modifications profoundly impact protein interactions, rendering the protein inactive in typical signaling pathways and serving as a marker of chronic oxidative stress within cells.

To validate the specificity and suitability of this antibody in our experimental setup, we have attached additional data confirming its performance (Fig. R2). We treated HCC1954 cells with 100 μM H_2O_2 for 4 hours, effectively inducing oxidative stress. This treatment resulted in detectable expression of oxidized PTPs, confirming the antibody's reliability and precision in our assays.

Fig. R2

Fig. R2 oxPTP expression in HCC1954 cells treated with H_2O_2 .

Expression of oxPTP in HCC1954 cells, which were treated with 100 μM H_2O_2 for 4 hours, stained for oxPTP, and assessed by flow cytometry. Live, single cells were gated for analysis. oxPTP expression levels are represented by the orange histograms, with unstained cells shown in blue.

8. $\text{O}_2^{\cdot-}$ should have a subscript (Introduction).

We have revised the manuscript to include the correct subscript notation.

Reviewer #4 (CyTOF, immune cell biology) (Remarks to the Author):

The manuscript by Wang et al. presents an innovative approach, termed single-cell reactive oxygen species (ROS) regulome profiling (scROP) to map the redox dynamics in immune cells at the single-cell level. The study is well-conceived and addresses a significant gap in understanding how redox states influence immune cell function, with relevant examples in the context of T cell activation, CAR-T cell persistence, and the effects of hypoxic conditions. However, there are a few areas where clarification, additional data, or methodological adjustments are necessary to strengthen the conclusions drawn.

Overall, I believe that the manuscript is a strong contribution to the field, offering a novel technique with significant potential. Addressing the comments below will help to clarify the study's impact and ensure that the findings are both reliable and broadly applicable.

Thank you for your thoughtful review of our manuscript and for recognizing the significance of our work. We appreciate your positive feedback and are grateful for your constructive suggestions.

Major points:

ROS modeling with H₂O₂ and ROS indicators (mitosox, H₂DCFDA, BODIPY):
The manuscript starts with a large-scale effort in identifying the most suitable antibodies for determining single-cell ROS states. While MitoSOX, H₂DCFDA, and BODIPY are standard indicators, their correlation with H₂O₂ concentration and duration needs to be better substantiated with data. A comparison of how these indicators correlate with each other under identical conditions would strengthen the methodology section, ensuring that these tools accurately reflect the ROS conditions in cells.

In our study, we included data from the SM826 cell line for three ROS indicators — H₂DCFDA, MitoSOX, and BODIPY. Under 100 μM H₂O₂ treatment for 48 hours, all three indicators showed a significant increase in fluorescence, demonstrating their reliability as measures of ROS levels in response to oxidative stress as shown in new Supplementary Fig. 4. We have clarified this in the text (lines 282-284): *“In contrast, traditional ROS indicators offer only generalized measurements of ROS levels (Supplementary Fig. 4).”*

Supplementary Fig. 4. Histogram of H₂O₂ treatment effects on ROS indicators in SM826 cell Line

Histograms illustrating the response of the SM826 cell line to H₂O₂ treatment across three redox indicators: left (MitoSOX), middle (H₂DCFDA), and right (BODIPY). The treatment conditions include different time points (0, 0.5, 4, and 48 hours) and concentrations (0 μM, 10 μM, and 100 μM) of H₂O₂. A black dashed line indicates the peak of the untreated group (0 h, 0 μM of H₂O₂) ($n=2$ independent samples, data for representative samples are shown).

Selection of ROS markers and clones:

Related to the above point, I think this manuscript could be a comprehensive resource for others if it were to include more antibody QC data. The selection of the final 25 ROS markers seems somewhat arbitrary without sufficient explanation. Including a more comprehensive rationale for selecting these markers, along with data for all tested clones, would provide transparency and allow readers to understand the decision-making process.

We have added a detailed explanation to the text and have now incorporated the 75 scatter plot QC results into Supplementary Table 1. The added explanation outlines the systematic and rigorous screening process we employed (lines 269-275): *“To select the most relevant antibodies for our ROS regulome panel, we first filtered out the antibodies that did not showed moderate correlations (correlation coefficient >0.3 or <-0.3) with any of the ROS-sensitive dyes under any conditions (Supplementary Fig. 1b). We then grouped the remaining 75 antibodies into seven modules based on their correlations (Supplementary Fig. 1c). Within each module, we calculated a weighted average to assess the importance of each antibody and ranked the antibodies by R values.”*

c

- up-regulated redox homeostasis
- ER disulfide bond formation
- membrane ROS-generating
- down-regulated redox homeostasis
- signal transduction and scavenger receptor
- cellular antioxidant mechanisms
- lipid and amino acid metabolism

Supplementary Fig. 1c. Heatmap of Spearman correlation coefficients between antibodies and three ROS indicators across six immune cell types, with data from 12 different H₂O₂ conditions. All values were transformed using the ASINH function. Antibodies were grouped into seven modules based on their correlation patterns: red, up-regulated redox homeostasis; orange, disulfide bond formation in the endoplasmic reticulum; yellow, membrane ROS-generating; blue, down-regulated redox homeostasis; dark blue, signal transduction and scavenger receptors; purple, cellular antioxidant mechanisms; and gray, lipid and amino acid metabolism. The color scale ranges from red (negative correlation) to blue (positive correlation).

scROP vs ROS indicators:

Including more data on the relationship between traditional ROS indicators and the scROP antibodies might also help to better demonstrate how scROP goes beyond the use of the existing indicators (which, one could argue, can already be used to determine single-cell ROS states).

Traditional ROS probe assays provide only a snapshot of overall ROS levels. In contrast, our scROP approach integrates the global redox status with molecular information through a multi-parametric assessment, enabling a more detailed comparison of redox activity across multiple cell types. This is clearly demonstrated in new Supplementary Fig. 2 and 3 (scROP) compared to new Supplementary Fig. 4, where scROP captures cell-type-specific and pathway-specific redox responses, whereas traditional ROS indicators provide only generalized measurements of ROS levels.

We have revised the text to clarify how scROP extends beyond the capabilities of existing indicators (lines 280-284): *“By simultaneously profiling over 25 ROS-related proteins, scROP captures cell-type-specific and pathway-specific redox responses (Supplementary Fig. 2 and 3). In contrast, traditional ROS indicators offer only generalized measurements of ROS levels (Supplementary Fig. 4).”*

T cell exhaustion versus activation:

The study's claim of T cell exhaustion in Figure 5 is questionable given the short time frame and in vitro conditions. T cell exhaustion typically requires chronic stimulation, which is difficult to replicate in vitro. The authors should consider reinterpreting these findings as prolonged activation rather than exhaustion unless additional functional data, potentially from in vivo models, can support the exhaustion claim. This would add credibility and depth to their conclusions.

To address this concern, we employed two well-established models of T cell stimulation: the acute and the chronic lymphocytic choriomeningitis virus (LCMV) infection models⁸. These models provide a nuanced understanding of T cell responses and exhaustion. In the acute LCMV model, mice are infected with a high dose of the LCMV Armstrong strain, leading to a robust and transient immune response. The virus is cleared rapidly, within 7-10 days, and T cell activation levels return to baseline. The chronic LCMV chronic model utilizes the LCMV clone 13 strain, which induces a prolonged and persistent infection that results in sustained T cell activation and eventual exhaustion, characterized by decreased functionality and increased expression of exhaustion markers.

On day 28 after chronic infection, CD8⁺ T cells have significantly higher levels of exhaustion markers, such as PD1 and TIM3, and lower levels of the precursor exhaustion marker TCF1/7 than do acutely infected T cells. This observation indicates that T cells are exhausted under chronic stimulation conditions. These models provide additional context for interpreting our observations and enhances the credibility of our conclusions regarding T cell exhaustion. These data are shown in new Fig. 5d and are described as follows in the text (lines 543-553): *“To validate these findings in a physiological context, we utilized the well-established in vivo T cell exhaustion model of chronic lymphocytic*

choriomeningitis virus (LCMV) infection. In the acute LCMV model, mice are infected with the LCMV Armstrong strain, leading to a robust and transient immune response. The virus is cleared rapidly, within 7-10 days, and T cell activation levels return to baseline. The chronic LCMV chronic model utilizes the LCMV clone 13 strain, which induces a prolonged and persistent infection that results in sustained T cell activation and eventual exhaustion. scROP analysis of the chronic LCMV model revealed redox signatures, characterized by elevated oxidative byproducts (oxPTP), diminished functional activity of SOD2(K68Ac), increased exhaustion markers (PD1 and TIM3), and reduced stemness markers (TCF1/7) over time, similar to those observed ex vivo (Fig. 5d and Supplementary Table 12).”

Fig. 5d. Histogram of average intensities of immune checkpoint inhibitors and ROS regulome markers in stimulated OT-I CD8⁺ T cells cultured with (dark blue) and without (light blue) N-AC at days 0, 3, and 5 ($n=2$, data from representative samples shown) and in T cells isolated at day 28 from acute (light green) and chronic (dark green) LCMV models ($n=3$, data from representative samples shown).

Detailed discussion of limitations:

The authors should include a more thorough discussion of the limitations of their study, including the challenges of interpreting redox data, the potential for variability in antibody-based detection, and the need for further validation in different biological contexts.

We agree that a discussion of limitations is needed and therefore we included the following in the revised manuscript (line 722-734): “Several limitations of this study should be acknowledged. Redox processes are inherently dynamic and context-dependent, which introduces challenges in capturing the full spectrum of oxidative stress responses. The redox state can be influenced by various factors, including environmental conditions, cell type, and metabolic status, contributing to variability in the observed results.

Additionally, although antibody-based detection methods like scROP offer specificity and sensitivity, variability can result from differences in antibody affinity, epitope accessibility, and cross-reactivity. Although rigorous validation of all antibodies used in our panel was conducted, further validation across different biological contexts will be necessary to generalize our findings and to enhance the applicability of the scROP assay. These limitations notwithstanding, our study lays the groundwork for future investigations into redox regulation, with the potential to significantly impact our understanding of disease mechanisms and therapeutic approaches.”

Minor points:

In Figure 1g, the manuscript should comment on the group of cells that appear to be of mixed lineages (left of basophiles). What are these cells and why were they grouped together?

We analyzed the cluster of cells that seem to represent mixed lineages in shown in the old Fig. 1g (new Fig. 1h). These cells are heterogeneous, which may be due to the presence of transitional states or to a combination of cell types with overlapping phenotypic characteristics. Importantly, in the mixed lineage cluster there are a higher proportion of central memory T cells and a lower proportion of effector memory T cells from both CD4⁺ and CD8⁺ populations than observed in T cell and non-mixed cell lineages.

The predominance of central memory T cells in this cluster suggests that these cells are in a state poised for long-term immune responses, capable of rapidly differentiating into effector cells upon re-encountering antigens. The reduced proportion of effector memory T cells indicates a low immediate response capability, which may reflect a transitional or activated state. This dynamic is further reflected in the UMAP analysis (Fig. 1h), where this cluster appears to be actively engaged, yet still maintains a reservoir of central memory T cells, highlighting the adaptability of T cell populations. This is now explained in the text (line 330-333): “Interestingly, we observed a group of cells composed of mixed lineages, which may be transitional cells that have a similar redox pattern after activation as indicated by the differential cellular composition of effector T cells and memory T cells (Supplementary Fig. 7).”

For Figure 2cde, the manuscript should clarify the number of donors shown (I guess it is one?) to ensure the robustness of the findings.

We now give the numbers of donors in these figure legends (line 819, 821, and 829). The updated legends now read: “*n=3 independent samples, data from one representative sample shown.*”

The rationale for selecting 15 metaclusters in Figure 3 should be clearly explained, including the criteria used for defining these clusters. In addition, there appears to be some inconsistency in terms of what is called a cluster vs. a metacluster.

These 15 clusters were defined based on the redox profiles of the cells, as determined through FlowSOM clustering analysis. The number of clusters was selected to optimally capture distinct cell populations while maintaining biological relevance. The "active" metacluster includes clusters 1, 2, 3, 4, 5, 6, 8, 11, and 13, whereas the "basal" metacluster consists of clusters 7, 9, 10, 12, 14, and 15 (Supplementary Fig. 10). These metaclusters were defined based on redox-related markers, such as oxPTP, PDI, and Ref/APE1, that effectively distinguish the active and basal states. To further highlight the differences between the basal and active metaclusters, we created a new figure, Fig. 3c, that illustrate the detailed scROP feature differences between the two states.

There is only a minor difference in pimonidazole staining between normoxic and hypoxic conditions in Supplementary Figure 6, which is surprising and should be discussed.

As shown in old Supplementary Fig. 7 (new Supplementary Fig. 13) we observed only a minor difference in pimonidazole staining between normoxic and hypoxic conditions. The heatmap provides an over view of the pimonidazole levels, which may obscure subtle differences in specific conditions. In new Supplementary Fig. 14, reproduced below, a histogram analysis shows a slight increase in pimonidazole expression under hypoxic conditions compared to normoxic conditions, suggesting that hypoxia does induce some level of response. It is important to note that although pimonidazole is a reliable marker for hypoxia, its staining intensity can be influenced by the overall oxidative environment.

Supplementary Fig. 14. Histogram of pimonidazole staining under hypoxia and normoxia conditions

Histogram of pimonidazole staining intensities in OT-1 T cells at day 2 under hypoxia (dashed light blue line), day 2 under normoxia (dashed dark blue line), day 4 under normoxia (solid light blue line), and day 4 under hypoxia (solid dark blue line). Data from one of two replicates are shown.

To contextualize these observations, previous studies have indicated that pimonidazole's effectiveness in detecting hypoxia can vary depending on cell type and metabolic profile. For instance, Wigerup et al. (2016) demonstrated that immune cells, including T cells, may sometimes show weaker or no labeling under hypoxic conditions, likely due to specific metabolic adaptations unique to these cells⁹.

The manuscript should offer a clearer explanation of what the CytoScore represents, including how it is calculated. The manuscript talks about localization, which I don't think is quantified here.

The CytoScore is a composite metric derived from the average expression levels of key redox-related markers. Specifically, CytoScore is calculated as the ASINH (inverse hyperbolic sine) transformed expression average of the following markers: HSP70, KEAP1, oxPTP, MTH1, and GR. This score provides an integrated measure of the cellular responses to oxidative stress, reflecting the overall antioxidant capacity within the cells. Similarly, the MitoScore is calculated as the ASINH transformed expression average of mitochondrial-specific markers: SOD2(K68Ac), oxDJ1, NNT, and GPX4. This score focuses on mitochondrial aspects of oxidative stress and the antioxidant responses that are critical to maintaining mitochondrial function. These scores do not quantify spatial localization of the markers but instead provide a functional perspective on oxidative stress and antioxidant responses within the cell. The mention of "localization" in the manuscript was intended to indicate the relevance of these markers to specific cellular compartments (such as the cytoplasm for CytoScore and mitochondria for MitoScore), rather than their physical distribution within the cells. We have revised the manuscript to provide a more detailed explanation of both CytoScore and MitoScore calculations and their relevance to the study (line 300-305): *"Specifically, CytoScore, which is a measure of the average expression of key redox markers within the cytoplasm, showed a strong correlation with H₂DCFDA intensities detected by flow cytometry (Fig. 1e, left panel). Similarly, the MitoScore, which is a measure of the average expression of mitochondrial-specific redox markers, correlated well with MitoSOX intensities measured by flow cytometry (Fig. 1e, middle panel)."*

References:

1. Heo, S., Kim, S. & Kang, D. The Role of Hydrogen Peroxide and Peroxiredoxins throughout the Cell Cycle. *Antioxidants* **9**, 280 (2020).
2. Lambeth, J.D. NOX enzymes and the biology of reactive oxygen. *Nature Reviews Immunology* **4**, 181-189 (2004).
3. La Flamme, A.C., Patton, E.A., Bauman, B. & Pearce, E.J. IL-4 plays a crucial role in regulating oxidative damage in the liver during schistosomiasis. *J Immunol* **166**, 1903-1911 (2001).
4. Peruzzo, R. et al. Exploiting pyocyanin to treat mitochondrial disease due to respiratory complex III dysfunction. *Nature Communications* **12**, 2103 (2021).
5. Levinthal, D.J. & Defranco, D.B. Reversible oxidation of ERK-directed protein phosphatases drives oxidative toxicity in neurons. *J Biol Chem* **280**, 5875-5883 (2005).
6. Son, Y. et al. Mitogen-Activated Protein Kinases and Reactive Oxygen Species: How Can ROS Activate MAPK Pathways? *J Signal Transduct* **2011**, 792639 (2011).
7. Taylor, M.F., Black, M.A., Hampton, M.B. & Ledgerwood, E.C. Insights into H₂O₂-induced signaling in Jurkat cells from analysis of gene expression. *Free Radic Res* **56**, 666-676 (2022).
8. Althaus, C.L., Ganusov, V.V. & De Boer, R.J. Dynamics of CD8⁺ T Cell Responses during Acute and Chronic Lymphocytic Choriomeningitis Virus Infection. *The Journal of Immunology* **179**, 2944-2951 (2007).
9. Vito, A., El-Sayes, N. & Mossman, K. Hypoxia-Driven Immune Escape in the Tumor Microenvironment. *Cells* **9**, 992 (2020).

Reviewer #3 (Remarks to the Author)

As noted in my original assessment, I consider this a sophisticated and powerful methodology, that with appropriate qualifications and cautions is a useful addition to the literature. It provides information on expression changes in cells treated with H₂O₂, and as a method for comparing treatments has the potential for further applications. However, I remain unconvinced that the pattern of changes can be taken as a specific indicator of an oxidative stress response. I reiterate that although a pattern of change may occur in response to an oxidant, the reverse conclusion that such a change is specifically due to an oxidant does not necessarily hold. In this respect the additional experiment with IL4 is helpful but does not overcome the problem. Therefore, for me to be satisfied, I would need a big change of emphasis in the context and conclusions. It would be a pity not to publish this extensive study, but in my opinion it needs to be heavily revised and presented in a much more moderate fashion, as a generalized signaling assay with some relationship to oxidative stress. I would not recommend it in its current context as a readout of a specific reactive oxygen species regulome.

Thank you very much for your thoughtful and constructive feedback. We truly appreciate your recognition of the potential of our methodology and your valuable suggestions for improving the manuscript. In response to your concerns, we have made significant revisions to the manuscript including the following major changes:

Removal of data collected using fluorescent probes: We agree that the use of ROS indicators did not serve as appropriate validation of our method. We now focus on broad signaling changes rather than implying a direct link to a specific reactive oxygen species.

Revision of terminology: We have made a fundamental shift in how we present our approach. The term "Single-cell reactive oxygen species regulome profiling" (scROP) has been replaced with "Single-cell Signaling Network under RedOx stress Profiling" (SN-ROP), which better reflects the nature of the assay. Additionally, the term "regulome" has been removed and replaced with "signaling network" to emphasize that the assay captures changes in the broad signaling landscape under redox stress rather than specific aspects of the ROS regulome.

Comprehensive revision: We have thoroughly revised the manuscript to demonstrate the value of our method as a generalized signaling assay that reflects oxidative stress responses. This revision is in line with your suggestion to moderate the emphasis and make the conclusions more balanced and cautious.

We believe that these changes strengthen the manuscript and provide a clearer description of our findings. We appreciate your helpful guidance in shaping the

manuscript to better reflect the broad applicability of this methodology to oxidative stress research.

I also still have major concerns about specific points from my original review.

Points 1 and 2. Use of fluorescent “ROS” probes. Despite the authors’ justification, information obtained from using these probes in the way they have done is essentially uninterpretable and meaningless. As I explained previously, I cannot see how making a single reading with an oxidant-sensitive probe added many hours or days after adding peroxide (which would be consumed in minutes) can be taken as a measure of ongoing oxidative stress. Understanding changes in response between adding the probe at 4 and 48 hours (new fig s4), and interpreting differences between cells in terms of oxidative stress are equally problematic. For a rigorous journal such as Nature Communications, these assays as they stand should not be used as a validation of their method and need to be removed from the submission.

Based on your feedback, we have removed the sections describing use of fluorescent ROS probes from the manuscript. We acknowledge that a single reading with an oxidant-sensitive probe after an extended period of peroxide exposure does not accurately reflect ongoing oxidative stress. Furthermore, we agree that interpreting differences in oxidative stress between 4-hour and 48-hour probe additions are problematic. Therefore, we have revised the manuscript to remove descriptions of these probe-based assays. We now focus on signaling network responses under redox stress as captured by our method, which we now call single-cell Signaling Network under RedOx stress Profiling (SN-ROP), a name that better reflects the nature of the assay. We have also removed discussions of specific ROS regulomes. We appreciate your guidance in improving the rigor of the experimental approach and the clarity of the manuscript.

Point 7. The authors have based their PTP oxidation findings on use of an antibody that apparently recognizes oxidation to the sulfonic acid. The apparent detection of PTPs oxidized to the sulfonic, (or sulfinic) acid under the conditions of this study is surprising as it would be expected to take a higher peroxide concentration exposure to the cells than used in this study (see for example Io Conte & Carrell JBC 2013). If any proteins formed sulfinic/sulfonic acids it would be peroxiredoxins, making the antiPrdxSO2 antibody a more effective detector. Was this the case?

We appreciate the reviewer’s insightful comments regarding the oxidation states of PTPs and the specificity of our detection approach. Although the presence of sulfinic or sulfonic acid of thiol enzymes is typically associated with high oxidative stress, a substantial

fraction of endogenous PTP1B was previously shown to be irreversibly oxidized in HepG2 cells (43%) and A431 cells (38%) without stimulation¹. These findings support the notion that PTP oxidation to the sulfonic and sulfinic acid states does occur under the experimental conditions established for our current study. Peroxiredoxins are highly susceptible to hyperoxidation, but our goal was not merely to select an effective oxidative detector but also to consider biological relevance. The ox-PTP antibody, which has been used to probe immune cells in prior studies², was selected because of its biological relevance.

Reviewer #4 (Remarks to the Author)

In their revised version of the manuscript, the authors have satisfied most of my initial comments. Importantly, they have included their antibody QC data, correlations with traditional ROS markers and new data to support their claim of T cell exhaustion. Some minor questions remain around the topics of antibody selection, correlation of ROS markers with each other (to show their robustness), and the reproducibility of some new data. Overall, I do think that the manuscript now provides an improved demonstration of the capabilities and limitations of the presented method.

We were pleased that the reviewer found that the revision better demonstrates the capabilities of our method, and we appreciate the constructive suggestions made by this reviewer.

Remaining comments:

The inclusion of the H₂O₂ treatment effects on ROS indicators and the heatmap of correlation values of these indicators with the tested antibodies helps to make their antibody selection process more transparent. However, I still do not quite understand how the 25 final markers were selected. Was this just the X top ranked antibodies in each correlation modules?

The final 25 markers were selected through a two-step process, focusing first on identifying antibodies with the greatest response to oxidative stress and then refining the selection based on their importance within the ROS-related signaling network. To filter out non-responding antibodies, the ASINH-transformed mean fluorescence intensity at 0 hours—averaged across three independent replicates—was used as the baseline for each antibody. Antibodies exhibiting more than a 10% deviation from this baseline under any condition (72 out of 103) were retained for further analysis. The 72 antibodies were

grouped into seven modules based on their co-regulation patterns. A weighted average score was calculated for each module, with larger modules (i.e., those with more antibodies) assigned a higher weight. This weighted score determined how many of the most responsive antibodies were selected, while maintaining an appropriate balance across the modules. This is now clearly stated in the Results section (SN-ROP: A multiplexed tool for single-cell analysis of redox-associated signaling) and in the Methods section (Antibody screening and validation).

Related, do I understand correctly that H₂DCFDA, BODIPY, and MitoSOX were measured simultaneously on the same cells? That would be important to also correlate these indicators with each other on a single cell level to assess how robustly these markers, which are here used as a gold-standard, reflect ROS conditions in the cells.

We appreciate your suggestion that we assess how H₂DCFDA, BODIPY, and MitoSOX correlate with each other and reflect ROS conditions in the cells. To clarify, we measured H₂DCFDA, BODIPY, and MitoSOX on different subsets of cells due to technical limitations in fluorescence detection and spectral overlap between the dyes. As a result, we divided the cell population into two groups: one group was treated with H₂DCFDA and BODIPY, and the other was treated with MitoSOX. This enabled accurate quantification of each ROS marker without interference between the dyes. In response to your comment, we have analyzed the correlations between the ROS indicators and found significant positive correlations as shown in Figure R1. These results suggest that the markers are correlated with each other, supporting their use as reliable readouts of ROS conditions within the cells, though with varying strengths of association. These findings suggest that these ROS indicators are accurate measures of oxidative stress.

Reviewer 3 was concerned that these oxidant-sensitive probes added hours or days after adding peroxide should not be used as validation for our method. Because of this reviewer's concerns, we have removed discussion of these fluorescent probes from our manuscript.

Fig. R1

Fig. R1. Pearson correlations of ROS indicators in activated OT-I CD8⁺ T cells

Pearson correlation analysis of ASINH-transformed expression levels across different activation states. Parallel samples were assessed using ROS indicators *via* flow cytometry. Left: Correlation between H₂DCFDA and MitoSOX intensities. Middle: Correlation between BODIPY and MitoSOX intensities. Right: Correlation between H₂DCFDA and BODIPY intensities. Circles indicate the mean population values of activated OT-I CD8⁺ T cells, color-coded by experimental day. Each data point represents triplicate measurements.

The inclusion of the LCMV exhaustion experiment is a worthy addition to the manuscript. I do want to point out though that the observed differences in the scROP markers are minute and on the edge of what I would consider detectable by CyTOF. It would be important to show that this difference can be robustly observed when looking at biological replicates.

We appreciate the reviewer's recognition of the importance of the LCMV exhaustion experiment. To further support the robustness of our findings, we have now included additional biological replicates in new Supplementary Figure 18 (reproduced below). Minor variations were detected between individual replicates, but the overall trend was highly consistent across all three biological replicates, demonstrating the reproducibility of our observations. Although the observed differences are near the detection limit of CyTOF, these changes were observed across independent experiments. The observation of consistent trends despite inherent biological variability strongly suggests that these differences are biologically meaningful rather than artifacts of technical noise. Furthermore, these findings align well with established mechanisms of T cell exhaustion, where oxidative stress adaptations often manifest as subtle but functionally significant shifts.

Supplementary Fig. 18. Expression of immune checkpoint inhibitors and ROS markers in LCMV models

Histogram of mean fluorescence intensities for immune checkpoint inhibitors and SN-ROP markers in LCMV acute (light green) and chronic (dark green) models at day 28 ($n=3$).

Related to Figure 2cde, I think that showing the pseudotime analysis for the other 2 donors (in the supplements) as was performed for the representative donor in the supplements would ensure consistency.

In response to this suggestion, we have included the pseudotime heatmaps for data from all donors in new Supplementary Figure 8 (reproduced below). This ensures that the analysis is presented comprehensively across all replicates.

Supplementary Fig. 8. Pseudotime analysis of protein expression in CD8⁺ T cells

(a)(c) Pseudotime values calculated using the SCORPIUS package plotted in a heatmap along with the 99th percentile normalized data, which was smoothed using a window size of 100.

(b)(d) Slope (first derivative) heatmap of protein expression across pseudotime. The vertical dashed lines indicate significant inflection points ($n=3$ independent samples, data from two samples shown).

References:

1. Lou, Y.-W. et al. Redox regulation of the protein tyrosine phosphatase PTP1B in cancer cells. *The FEBS Journal* **275**, 69-88 (2008).
2. Choi, S. et al. THEMIS enhances TCR signaling and enables positive selection by selective inhibition of the phosphatase SHP-1. *Nature Immunology* **18**, 433-441 (2017).